# Dynamic transcriptome and chromatin architecture in granulosa cells during chicken folliculogenesis

Diyan Li [1,3 ✉], Chunyou Ning[1,3], Jiaman Zhang[1,3], Yujie Wang [1,3], Qianzi Tang [1], Hua Kui [1], Tao Wang[1], Mengnan He[1], Long Jin [1], Jing Li[1], Yu Lin[1], Bo Zeng[1], Huadong Yin[1], Xiaoling Zhao[1], Yao Zhang[1], Huailiang Xu[2], Qing Zhu[1] & Mingzhou Li [1 ✉]

Folliculogenesis is a complex biological process involving a central oocyte and its surrounding somatic cells. Three-dimensional chromatin architecture is an important transcription regulator; however, little is known about its dynamics and role in transcriptional regulation of granulosa cells during chicken folliculogenesis. We investigate the transcriptomic dynamics of chicken granulosa cells over ten follicular stages and assess the chromatin architecture dynamics and how it influences gene expression in granulosa cells at three key stages: the prehierarchical small white follicles, the first largest preovulatory follicles, and the post-ovulatory follicles. Our results demonstrate the consistency between the global reprogramming of chromatin architecture and the transcriptomic divergence during folliculogenesis, providing ample evidence for compartmentalization rearrangement, variable organization of topologically associating domains, and rewiring of the long-range interaction between promoter and enhancers. These results provide key insights into avian reproductive biology and provide a foundational dataset for the future in-depth functional characterization of granulosa cells.

[1] Institute of Animal Genetics and Breeding, College of Animal Science and Technology, Sichuan Agricultural University, Chengdu 611130 Sichuan, China. [2] College of Life Science, Sichuan Agricultural University, Ya'an 625014 Sichuan, China. [3] These authors contributed equally: Diyan Li, Chunyou Ning, Jiaman Zhang, Yujie Wang. ✉email: diyanli@sicau.edu.cn; mingzhou.li@sicau.edu.cn

The ovary is a reproductive organ in vertebrates that consists of follicles at several different developmental stages. As the basic unit of reproduction, ovarian follicles are composed of a central oocyte and the surrounding endocrine cells (the inner layer is composed of granulosa cells (GCs) and the outer layer is composed of thecal cells). During folliculogenesis, oocytes undergo a complex regulatory process resulting from instructive paracrine and junctional interactions with GCs[1]. This relationship between oocytes and GCs allows for the exchange of regulatory signaling molecules that control oocyte meiosis, cell cycle progression, tissue morphogenesis, and cytoskeletal remodeling, all of which are important for folliculogenesis and oogenesis[2,3].

The domestic chicken (*Gallus gallus domesticus*), which includes broiler (meat-producing) and layer (egg-producing) chickens, is of enormous agricultural significance and represents a classic model to study folliculogenesis[4]. Ovarian follicles in chickens develop in a continuous and hierarchical process[5] that depends on the activation of the hypothalamic-pituitary-gonadal axis. When hens lay eggs, the functionally mature ovary contains hundreds of prehierarchical follicles, including small and large white follicles (SWFs and LWFs), small and large yellow follicles (SYFs and LYFs), 5–6 growing preovulatory follicles (demarcated by volume sequentially as F6 or F5, F4, F3, F2, and F1), and 2–4 postovulatory follicles (POFs) that are devoid of oocytes[6]. They enter the preovulatory hierarchy from the cohort of prehierarchical follicles daily (~6–8 mm in diameter), after which they are typically destined for ovulation[7]. Postovulatory follicles typically disappear within several days and do not form the corpus luteum in chickens[8,9]. As such, this normally rapid degradation is required for new hierarchical recruitment and subsequent ovulation[10].

Substantial research efforts have been made to characterize developmental alterations in the morphology and transcriptional regulation of GCs in birds, particularly during folliculogenesis, where there is a focus on the activation of primordial follicles or the selection of a dominant follicle[11–13]. In the eukaryotic cell nucleus, genomic DNA is highly folded and spatially organized into a hierarchy of 3D structures, including chromosome territories, compartments, topologically associating domains (TADs), and long-range interactions[14–17], which play important roles in transcriptional regulation[18]. Nevertheless, a comprehensive characterization of the developmental reprogramming of chromatin architecture associated with transcriptional regulation throughout follicle development has not yet been performed in birds.

In this study, we investigate the transcriptomic dynamics of chicken GCs in ovarian follicles across ten key developmental stages and generate high-resolution chromatin contact maps for GCs across three major developmental stages using in situ high-throughput chromatin conformation capture (Hi-C) sequencing. These experimental settings allowed us to conduct an integrated analysis of chromatin structure and transcriptomic characterization of chicken GCs associated with various physiological functions during folliculogenesis.

## Results

**Dynamic transcriptome in GCs during chicken folliculogenesis.** We first depicted developmental changes in the transcriptome of chicken GCs through ten key folliculogenesis stages (four prehierarchical [SWF, LWF, SYF, and LYF], five preovulatory [F5, F4, F3, F2, and F1], and one postovulatory [POF]) (Fig. 1a). We then generated a total of 812.45 Gb high-quality bulk RNA-seq data with six biological replicates for each stage (~13.54 Gb sequences per sample) (Supplementary Data 1). A total of 14,418 genes (85.92% of the annotated genes in the genome) had evident expression (transcripts per million

[TPM] > 0.5 in at least three replicates for a given stage) during the folliculogenesis, which typically had a developmental stage-dependent pattern and is highly reproducible within biological replicates (Spearman's $r > 0.80$) (Supplementary Fig. 1a, b).

Only half (88 genes, or ~52.38%) of the most abundant genes (the top 1%; 168 genes) for a given stage were shared throughout ten stages (Supplementary Fig. 1c, Supplementary Data 2). These most abundant genes were commonly involved in metabolic and cell adhesion processes such as peptide biosynthetic process, cytoplasmic translation, and adherens junction, and also specifically involved in reproductive, signaling, and localization processes such as female gamete generation for the SWF stage, the regulation of intrinsic apoptotic signaling pathway for the SYF and LYF stages, and glucocorticoid response for the F1 stage (Supplementary Fig. 1d). These results highlight the differences in biological functions occurring in GCs during stepwise folliculogenesis. Additionally, we performed pairwise differential expression analysis for the 10 developmental stages (Supplementary Fig. 1e) and found that of the four prehierarchical stages, the comparison between SWF and POF had the greatest number of differentially expressed genes (DEGs). For preovulatory follicles, gene expression in the F1 stage most differed from the POF stage. As such, we used SWF, F1, and POF to represent the transcriptional features at the prehierarchical, preovulatory, and postovulatory stages, respectively.

We next investigated possible developmental scenarios for gene expression throughout folliculogenesis and identified four main patterns involving 3669 genes using the maSigPro-GLM algorithm[19](Fig. 1b, Supplementary Data 3). We found that a total of 3172 genes (1632 and 1540 genes in clusters 1 and 2, respectively) were typically upregulated during prehierarchical stages and that genes in cluster 1 are specifically upregulated during the SWF stage (Fig. 1b). Functional enrichment analysis using Metascape suggested that these genes are primarily involved in cell cycle and gametogenesis processes such as cell division, gamete generation, and spindle localization (Fig. 1c). These likely reflect the proliferation and development of GCs in the small white follicles. In contrast, a total of 497 genes (171 and 326 genes in clusters 3 and 4, respectively) were typically upregulated during preovulatory stages. Of these, the genes in cluster 4 exhibited higher expression levels at the postovulatory stage than other stages (Fig. 1b). The genes in cluster 3 were primarily involved in growth and development processes, indicating that the number of GCs responding to follicular enlargement in the first large preovulatory follicles rapidly increased. Genes in cluster 4 were mainly related to autophagy and proteolysis and were associated with the degeneration of ovarian follicles caused by the apoptosis of GCs in POF (Fig. 1c). This indicates that dynamical gene expression in a developmental stage-dependent manner is related to functional divergences during folliculogenesis from the prehierarchical to preovulatory and postovulatory stages.

To accurately depict transcriptomic changes in GCs during folliculogenesis, we used a single-cell transcriptome (scRNA-seq) approach to dissect the transcriptional differences among three representative stages on a 10× Genomics system: the SWF, F1 and POF stages. After quality filtering, the transcriptome profiles of 21,393 cells were available for cell-type characterization (6596, 5996, and 8801 cells for SWF, F1, and POF, respectively) (Supplementary Data 1). To confirm the identity of these GCs, we used publicly available transcriptome profiles of 22,561 cells derived from chicken hearts[20] as comparative controls. We performed the uniform manifold approximation and projection analysis and surveyed the expression patterns of the top 50 most variable genes (Supplementary Fig. 2a, b). We then colored the single cells according to the expression levels of five canonical markers of GCs (*CYP11A1*[21], *CHST8*[22], *FSHR*[22,23], *TSPAN6*, and

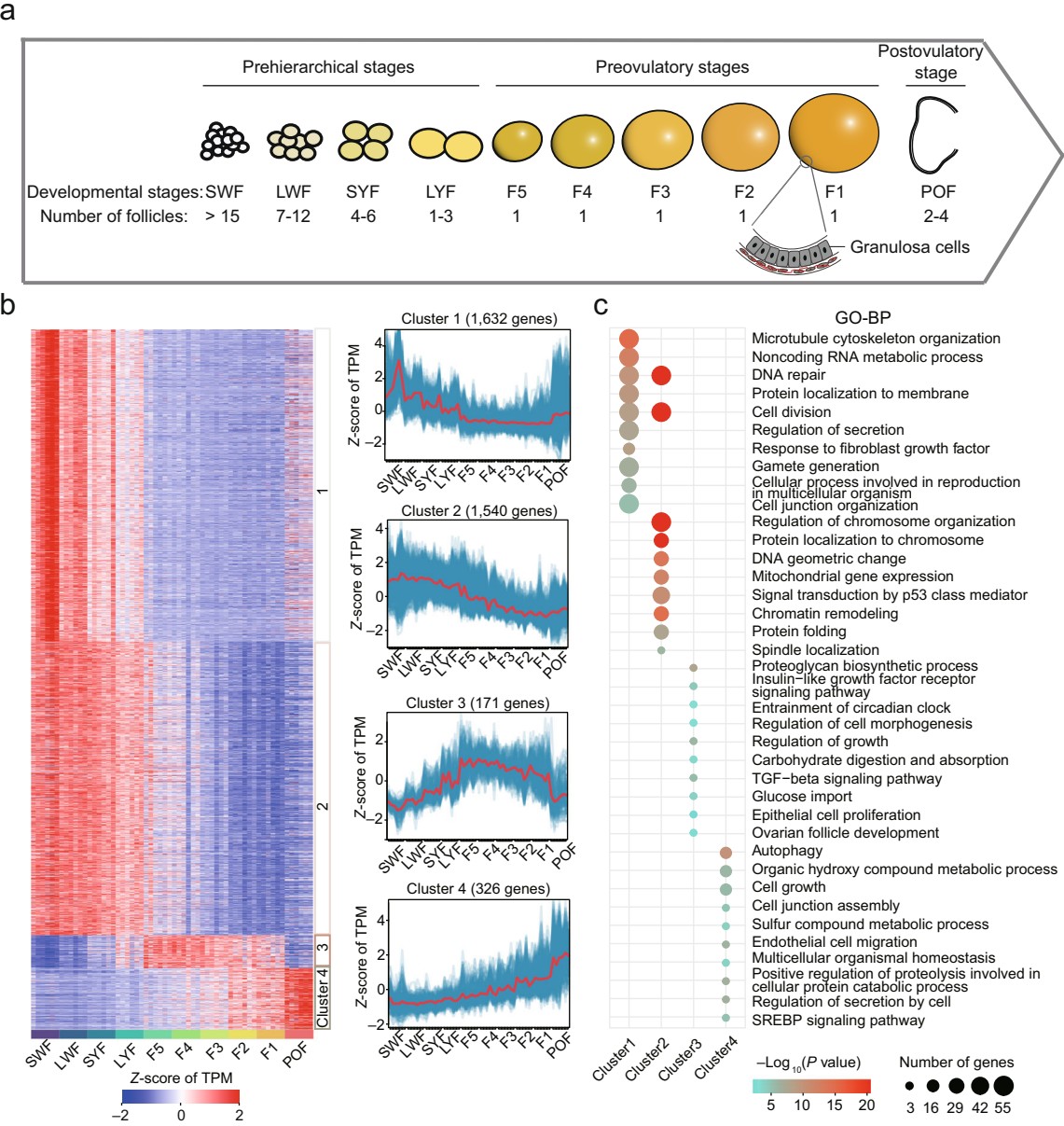

**Fig. 1 The transcriptomic profiles of granulosa cells (GCs) during folliculogenesis. a** Schematics of chicken ovarian follicle development at ten time points during folliculogenesis. The number of follicles for each stage in the chicken ovary is indicated below the stage. GCs in the F1 follicle are indicated on the plot. **b** Expression profiles of four temporal expression clusters revealed by *k*-means clustering. Left: Expression heatmap drawn using *Z*-score of TPM values for each gene in the four clusters. Right: Temporal expression profiles of the four clusters. The red lines represent mean gene expression levels, and the blue lines represent gene expression levels for each gene in the relative cluster during folliculogenesis. **c** The top ten significantly enriched Gene Ontology-Biological Process (GO-BP) terms for genes in each cluster. Source data are provided as a Source Data file.

*DSP*[24]) and five representative differentially expressed genes (*NOV*, *RLN3*, *EDN2*, *FGL2*, and *RGS16*) by the GCs detected in this study (Supplementary Fig. 2c, d). We found the vast majority of collected cells (21,336 of 21,393, or 99.73%) possess characteristics typical of GCs (clusters 0, 3, and 4) (Supplementary Fig. 2a, b). This ensures that the cell purity of the GCs was maintained.

The unique expression profiles of GC genes across three stages (Supplementary Fig. 2e) confirmed that over-representative functions of stage-specifically expressed genes align with specific physiological changes during folliculogenesis. Genes that were specifically expressed in GCs at the SWF stage were primarily involved in the cell division cycle, while the GCs reflected in SWFs typically grow and are activated when external hormones

are stimulated. Genes exclusively expressed at the F1 stage were mainly involved in the biosynthetic processes of glycolipid and secretion regulation, and generally match the physiological functions of F1 follicles. After ovulation, the GCs begin to regress via apoptosis and inflammatory responses, producing genes related to the above functions that were specifically expressed in GCs at the POF stage (Supplementary Fig. 2f). We also identified a subset of signature genes for GCs that exhibited expression changes across the three stages (Supplementary Data 4). These have potential as stage-specific GC markers. For example, the anti-Müllerian hormone (*AMH*), a typical marker in human GCs[25], was preferentially expressed in GCs at SWFs in chickens (Supplementary Fig. 2g). This highlights a significant difference in folliculogenesis between mammals and birds.

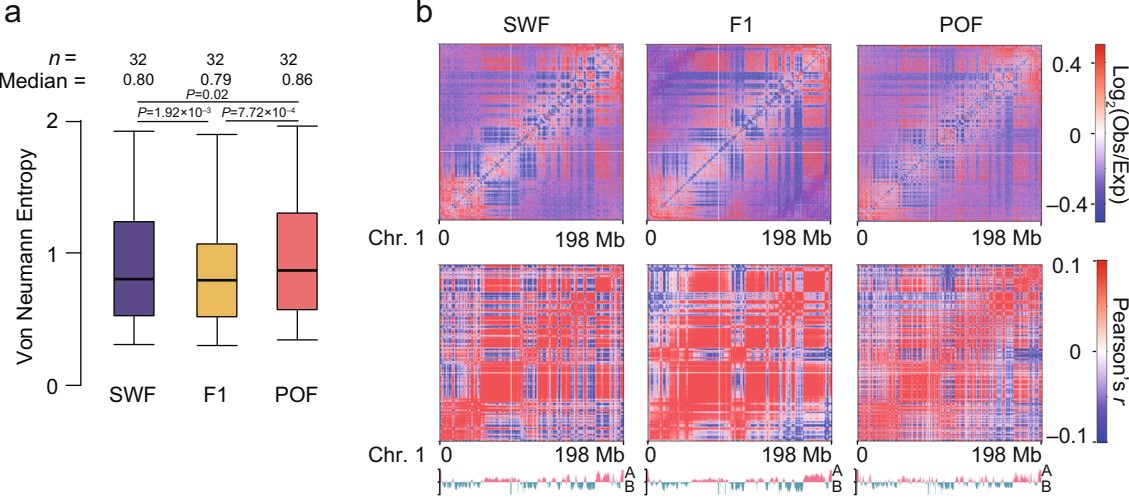

**Fig. 2 Dynamic changes of chromatin architecture during granulosa cell development. a** Quantification of the disorder in the chromatin structure of the whole genome using Von Neumann Entropy (VNE). *P* values were calculated using two-sided Wilcoxon rank-sum test. In the boxplot, the internal line indicates the median, the box limits indicate the upper and lower quartiles and the whiskers extend to 1.5 IQR from the quartiles. **b** Normalized Observed/Expected contact maps (top panels) and correlation matrixes of PC1 values (bottom panels) at 20 kb resolution for chromosome 1. Source data are provided as a Source Data file.

**Developmental changes in compartmentalization and local accessibility for GCs**. To elucidate the multiscale rewiring of chromatin architecture and its influence on GCs gene expression during folliculogenesis, we used in situ Hi-C to map chromatin contacts for GCs across SWF, F1, and POF. We generated a total of ~2.24 billion valid contacts (~373.77 million [M] contacts per sample (Supplementary Tables 1 and 2) and reached a maximum resolution of 5 kb by merging the intrachromosomal contacts of the replicates at each stage) (Supplementary Fig. 3a, Supplementary Tables 3 and 4). Most (~66.21%) contacts occurred within chromosomes, exhibited high reproducibility among the biological replicates (Supplementary Fig. 3b–e), and consisted of the dominant (~64.97%) long-range interactions (≥20 kb) (Supplementary Fig. 3f). All samples showed a strong decrease in contact probability with an increase in the distance between loci (Supplementary Fig. 3g).

Next, we calculated multivariate Von Neumann Entropy (VNE)[26] to measure the changes in 3D structural order during folliculogenesis. We observed a significantly higher VNE in the POF stage (0.86, $P < 0.016$, Wilcoxon rank-sum test) than in the SWF (0.80) and F1 stages (0.79) (Fig. 2a). This is likely due to a more disordered and relaxed chromatin architecture in the POF stage (Fig. 2b), while the architecture is more stable and ordered in mature GCs at the F1 stages, which aligns with the relaxed genome architecture observed during senescence[27].

At the sub-chromosome level, we explored various compartmental rearrangement scenarios during folliculogenesis and observed correlations between primary transcription features and the chromatin compartmental status. Replicates of each stage shared similar A/B compartment patterns and ~49.62% of the whole genome were Compartment A bins (Supplementary Fig. 4a, b). Compartment A was positively correlated with Guanine-Cytosine content (Spearman's $r > 0.30$, $P < 2.20 \times 10^{-16}$) (Supplementary Fig. 4c), and gene expression levels in Compartment A were significantly higher than in Compartment B ($P < 2.20 \times 10^{-16}$, Wilcoxon rank-sum test) (Supplementary Fig. 4d). The spatial organization of the compartments constructed by miniMDS[28] showed a negative correlation between the PC1 value / gene expression and distance from the center of the nucleus (i.e., nuclear radius) (Supplementary Fig. 4e). This is consistent with the spatial location preferences of euchromatin

and heterochromatin[29] and is similar to the structure found in mammals[30].

We identified substantial levels of compartmental switching in GCs across three stages (~354.50 Mb, or ~36.90% of the genome) (Fig. 3a, b; Supplementary Fig. 5a, b). In these regions, most switching was unidirectional (63.53%) (Fig. 3b, c): from A to B ("AAB" and "ABB", 130.80 Mb) or from B to A ("BAA" and "BBA", 94.42 Mb). The rest were transient switches, either "ABA" or "BAB" (129.28 Mb) (Fig. 3b, c). Although we observed dynamic compartmentalization during folliculogenesis, these were accompanied by slight changes in gene expression, significant differences in gene expression undergoing compartmental switching have only been observed at the SWF and POF stages (Fig. 3d). This result suggests the finitely contribution of the rearrangement of compartmentalization on the alterations of gene expression (measured as the mRNA abundances) during folliculogenesis, which accordance with the observations during stem cell differentiation[31]. The relatively weak correlation between changes in compartmentalization and gene expression is most likely due to the relatively steady-state mRNA abundances determined using bulk RNA-seq, thus further analysis of the nascent transcription (such as Bru-seq [bromouridine labeling and sequencing] approach)[32] is required to identify this regulatory mechanism.

Functional enrichment analysis demonstrated that genes embedded in regions experience the A-to-B switching event and were primarily involved in hormone activity and signaling processes (Supplementary Fig. 5c). This includes *AMH* (regulating the preselection pool), *MOS* (maintaining the ordered reproduction process), *FGF20*[33], and *FMNL2*[34] (which is involved in the regulation of cell growth) (Supplementary Fig. 5d, Fig. 3e). Nonetheless, genes located in regions that were subject to B-to-A switching events were primarily involved in stimulus-response, developmental, and immune system processes (Supplementary Fig. 5c). This includes *NFATC1*[35], which reflects the autophagy activities that occur during follicle degeneration. Moreover, the genes essential for follicle maturation, such as *FABP6*[36], *STAR*[37] (which is related to sterol synthesis), and *LHCGR*[38] (encoding luteinizing hormone), were specifically located in the compartment A region at the F1 stage (Supplementary Fig. 5d, Fig. 3e).

We further performed an ATAC-seq assay to measure the differences in local accessibility during folliculogenesis

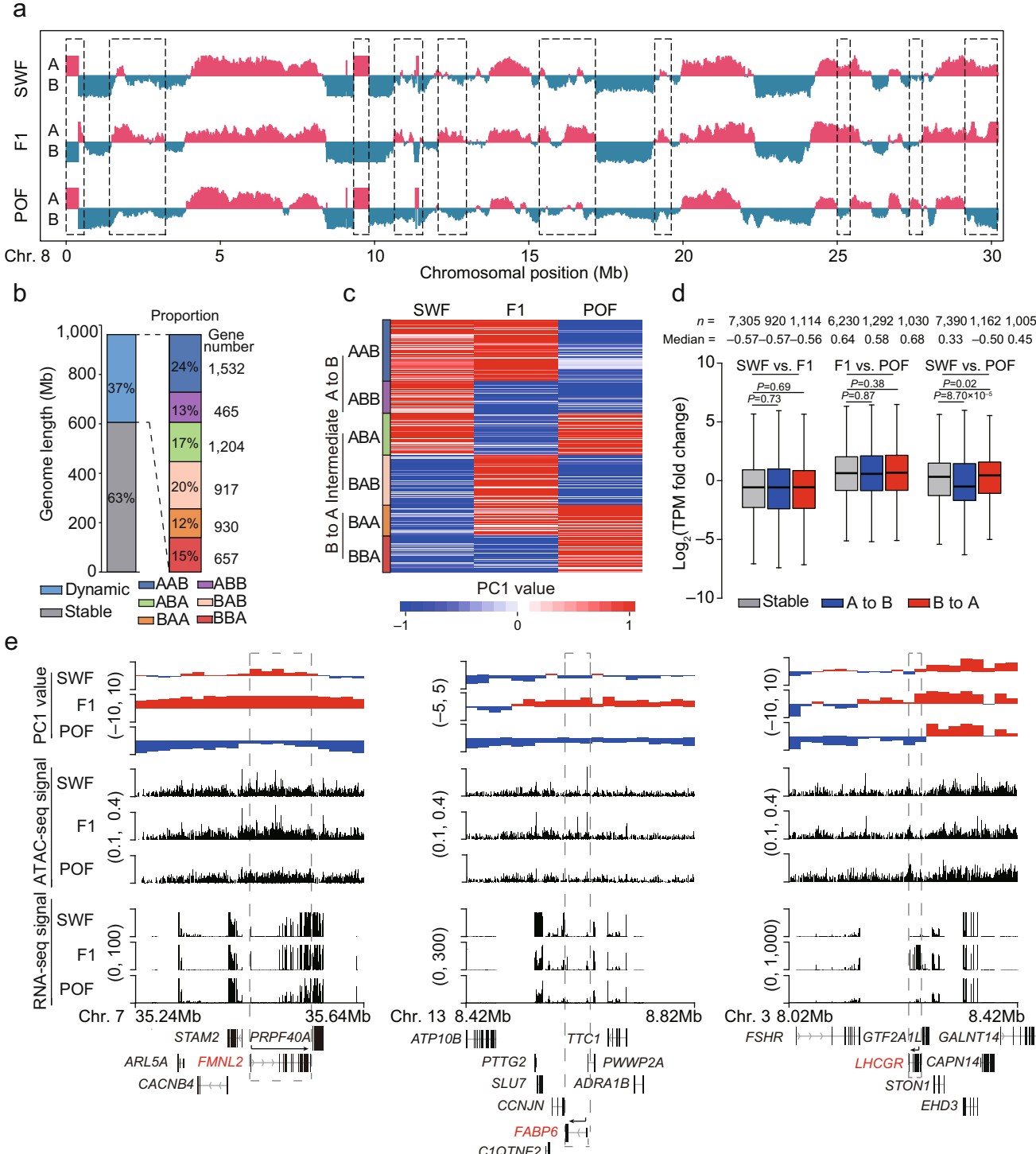

**Fig. 3 Compartmentalization dynamics in chicken granulosa cells during folliculogenesis. a** Compartment switches (gray dotted boxes) on chromosome 8. A/B compartments are indicated by PC1 values. **b** Genomic lengths and proportions of stable and dynamic compartments. Dynamic compartments are classified into six types of transitions. **c** Heatmap of the PC1 values for the compartment switching regions. **d** Expression levels of DEGs located in compartment switching regions (from A to B: blue; from B to A: red) compared to those in stable compartments (gray). $P$ values were calculated using two-sided Wilcoxon rank-sum test. In the boxplot, the internal line indicates the median, the box limits indicate the upper and lower quartiles and the whiskers extend to 1.5 IQR from the quartiles. **e** Three representative functional genes (red) subject to compartment switching during folliculogenesis, including *FMNL2*, *FABP6*, and *LHCGR*. The dashed line boxes indicate the chromosomal locations of the interested genes. The tracks show the compartment (top panels), ATAC (middle panels), and gene expression (bottom panels) features within 400 kb. Gene structures are indicated below the tracks. The black arrows indicate the direction of the gene transcription. Source data are provided as a Source Data file.

(Supplementary Data 1, Supplementary Fig. 6a). As expected, we found that the A compartments were enriched with more ATAC-seq signals than B compartments (Supplementary Fig. 6b, c). Therefore, the A compartments are more accessible. We observed stage-specific peaks in the GCs at the SWF and F1 stages, which are enriched in motifs corresponding to the transcription factors (TFs) in the GATA family (Supplementary Fig. 6d). These are essential for development, differentiation, and homeostasis[39,40], suggesting the importance of these TFs in SWF and F1 stages. In contrast, POF-specific peaks in GCs are enriched in motifs corresponding to TFs involved in cytotoxicity and apoptosis induction[41–43], including KLF5, PITX1, and OTX1 (Supplementary Fig. 6d). These results support the physiological course of chicken folliculogenesis, provide evidence supporting that chromatin state-mediated compartment activation is associated with transcriptional regulation, and directly implicate multiple loci that exhibited distinct compartmentalization and accessibility during folliculogenesis.

**Most TADs in GCs were highly stable during folliculogenesis.** At the submegabase scale, the local chromatin architecture can be characterized by TAD. To investigate the conservation of TAD in different cells, we downloaded chicken fibroblast and erythrocytes Hi-C data (including immature and mature erythrocytes) and performed comparative analysis. We used both the directionality index (DI)[44] and the insulation score (IS)[45] methods to identify TAD structures at 20 kb resolution. The TAD boundaries produced by the two algorithms were highly reproducible within GC biological replicates (Spearman's $r$ (DI) > 0.94, Spearman's $r$ (IS) > 0.97) (Fig. 4a, Supplementary Fig. 7a) and were divergent from other cell types, particularly the immature and mature erythrocytes (Supplementary Fig. 7b). As expected, TAD boundaries were enriched for the TSS of protein-coding genes, especially for housekeeping genes (Supplementary Fig. 7c). We found that ~98.30% of TAD boundaries were invariant in GCs during folliculogenesis (Fig. 4b). There were 1,996, 1,361, and 1,326 TAD boundaries identified in fibroblasts cells, immature and mature erythrocytes, respectively; and 1,637 (82.01%), 760 (55.84%), and 752 (56.71%) of the boundaries were conserved between GCs and them, respectively (Supplementary Fig. 7d, e). A total of ~1,900–2,000 TADs were subsequently detected in chicken GCs, with a median size of ~400 kb and occupying ~86.58% of the length of the genome (Fig. 4c, Supplementary Fig. 7f). We found that TADs enriched by A compartments (A TADs were defined as having over 70% TAD bins belonging to A compartment) were generally smaller in size than TADs primarily located in B compartments (B TADs) (Fig. 4d). This could be due to local architecture accommodating more active gene regulation by local interactions in gene-dense regions.

We observed a stronger TAD boundary insulation at the F1 stage than that in SWF and POF, indicating increased spatial segregation of local chromatin when the follicle reached maturity (Fig. 4e, f, Supplementary Fig. 8). This was consistent with most orderly chromatin spatial organization found in this stage (Fig. 2a, b). A small portion (1.43–1.98%) of TAD boundaries were gained or lost in a stage-specific manner during folliculogenesis (Fig. 4b, Supplementary Fig. 9a, b). These stage-specific boundary genes were primarily implicated in the developmental growth and regulation of cellular response to stress processes (Supplementary Fig. 9c). Analyzing gene expression at dynamic boundaries demonstrated that a boundary gain was associated with upregulated gene expression, while a boundary loss was associated with downregulated gene expression (Supplementary Fig. 9d). For example, ZEB2 regulates cell-cell adhesion and rearrangements of cytoskeletal architecture by mediating

E-cadherin expression[46], and when it gained a TAD boundary its expression increased at the F1 stage (Fig. 4g). Overall, most TADs identified in GCs were stable during folliculogenesis.

**Fluctuating intra-TAD interactions associated with transcription changes during folliculogenesis.** Studies have demonstrated that the genomic positioning of the TAD structure can remain stable[44,47], yet the frequency of interactions within TAD varies with domain-wide changes during the differentiation and reprogramming processes[31,48]. To assess the kinetics of intra-TAD interactions during follicle development, we assessed 1831 consensus TADs (cTADs, take up 87.40% of the genome with a median size of ~420 kb) across various stages and replicates (Fig. 4c) and calculated the domain score (D-score, defined as the fraction of intra-TAD contacts over the total intrachromosomal contacts[49] and reflects self-interactive tendencies within a domain) of cTADs for each sample (Supplementary Fig. 10a). As previously reported[49], the cTADs with higher D-score values were preferentially accessible and had higher levels of gene expression (Fig. 5a, b).

After analyzing the D-score changes between consecutive stages, we found that a total of 174–342 cTADs (~85.54–176.42 Mb) exhibited significant alterations ($P < 0.05$, Student's $t$ test) in intra-TAD interaction frequency during folliculogenesis; 9.50% of cTADs changed between the SWF and F1 stages and 18.68% of cTADs changed between the F1 and POF stages (Supplementary Fig. 10b). Compared with unchanged TADs, the TADs with decreasing D-scores had significantly reduced gene expression levels, while TADs with increasing D-scores showed slight but not statistically significant increases in gene expression levels (Fig. 5c). Functional enrichment analysis indicated that the D-score changes aligned with typical GC biological functions during folliculogenesis (Supplementary Fig. 10c). For example, CDH2 encodes a classical cadherin that forms adherens junctions between oocytes and GCs[50] had both a higher level of gene expression and a greater D-score at the F1 stage (Fig. 5d). These results suggest that folliculogenesis in chickens is accompanied by intra-TAD changes in GCs.

**Global rewiring of PEIs underpinning functional divergence in GCs during folliculogenesis.** The interactions between enhancers and their target-gene promoters (PEIs) are an important part of the gene regulatory process and could be causally related to spatiotemporal expression. Therefore, we next compiled an extensive genome-wide catalog of PEIs (median size of ~85 kb that primarily existed in TADs [67.49%]) in GCs (Supplementary Fig. 11a, b) for each stage at a 5 kb resolution (Supplementary Table 4) using the PSYCHIC algorithm[51]. PEIs cannot be reliably inferred from a genomic distance, therefore, we observed that ~53.82% of enhancers interacted with a more distant promoter instead of those close by (Supplementary Fig. 11c). This spatial proximity data highlights the complexity of PEIs[52,53].

To demonstrate how extensive PEI rewiring contributes to transcriptomic divergence in GCs during folliculogenesis, we used a regulatory potential score (RPS) to measure the regulatory effects of multiple enhancers for a given gene. As expected, genes with larger RPS had higher levels of expression (Fig. 6a), which confirmed the additive effects of enhancers can increase gene expression (Supplementary Fig. 11d).

To further explore the role of enhancers in transcription control, we measured the activities of the putative enhancers involved in PEIs by analyzing the distribution of H3K27ac, which is a typical histone marker of active enhancers (Supplementary Fig. 11e). This allowed us to compare the differences in how enhancers and promoters affect transcriptional activity[54–56]. Our results demonstrated that ~20.04% PEI-associated genes

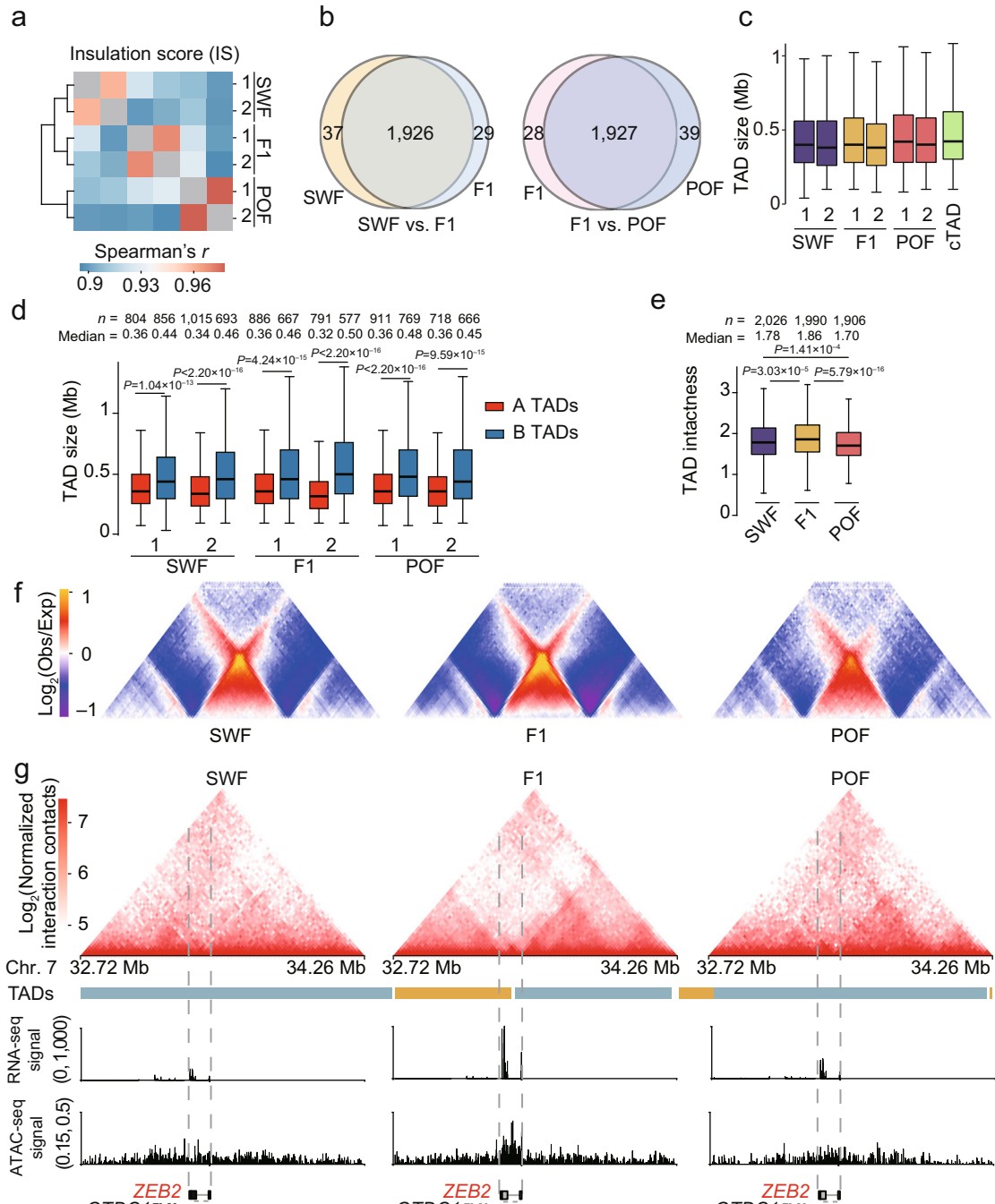

**Fig. 4 TAD boundaries are largely stable in chicken granulosa cells during folliculogenesis. a** Spearman's *r* heatmap of the insulation score (IS) at different developmental stages. **b** Overlap of TAD boundaries between successive stages of follicle development. **c** Boxplots showing TAD sizes in GCs during folliculogenesis, where cTADs represent consensus TADs. **d** Comparison of TAD size for A and B TADs. **e** TAD intactness at each stage. **f** Examples of average TAD representation with intra- or inter-TAD contact in SWF, F1 and POF respectively. **g** Changes in TAD boundaries and expression levels at the locus of *ZEB2* gene (red) across different developmental stages. Top: Hi-C contact heatmaps of the genomic region around *ZEB2* (Chr.7: 32.72–34.26 Mb). Middle: TAD boundaries and genome browser tracks of gene expression and ATAC-seq signals. Bottom: gene structures in the genomic region. The dashed line boxes indicate the chromosomal locations of the genes. For **c**, **d** and **e**, the internal line indicates the median, the box limits indicate the upper and lower quartiles and the whiskers extend to 1.5 IQR from the quartiles. *P* values in **d** and **e** were calculated using two-sided Wilcoxon rank-sum test. Source data are provided as a Source Data file.

contacted with super-enhancers (SEs, with broad H3K27ac signal) exhibited higher RPS and had increased expression than genes (~46.29%) contacted with regular enhancers (REs, with moderately H3K27ac signal) or genes (~33.67%) contacted with poised enhancers (PEs, depleted in H3K27ac signal) (Supplementary Fig. 11f–h).

We next identified six representative developmental patterns for 2659 genes that exhibited differential RPS between stages (|$\log_2$FC| >1.5 and |Δ| > 3) and were accompanied by changes in enhancer activity (Fig. 6b, c). The genes exhibited preferential RPS values in a developmental stage-dependent manner (Fig. 6c, Supplementary Fig. 11i), which involved a distinct functional

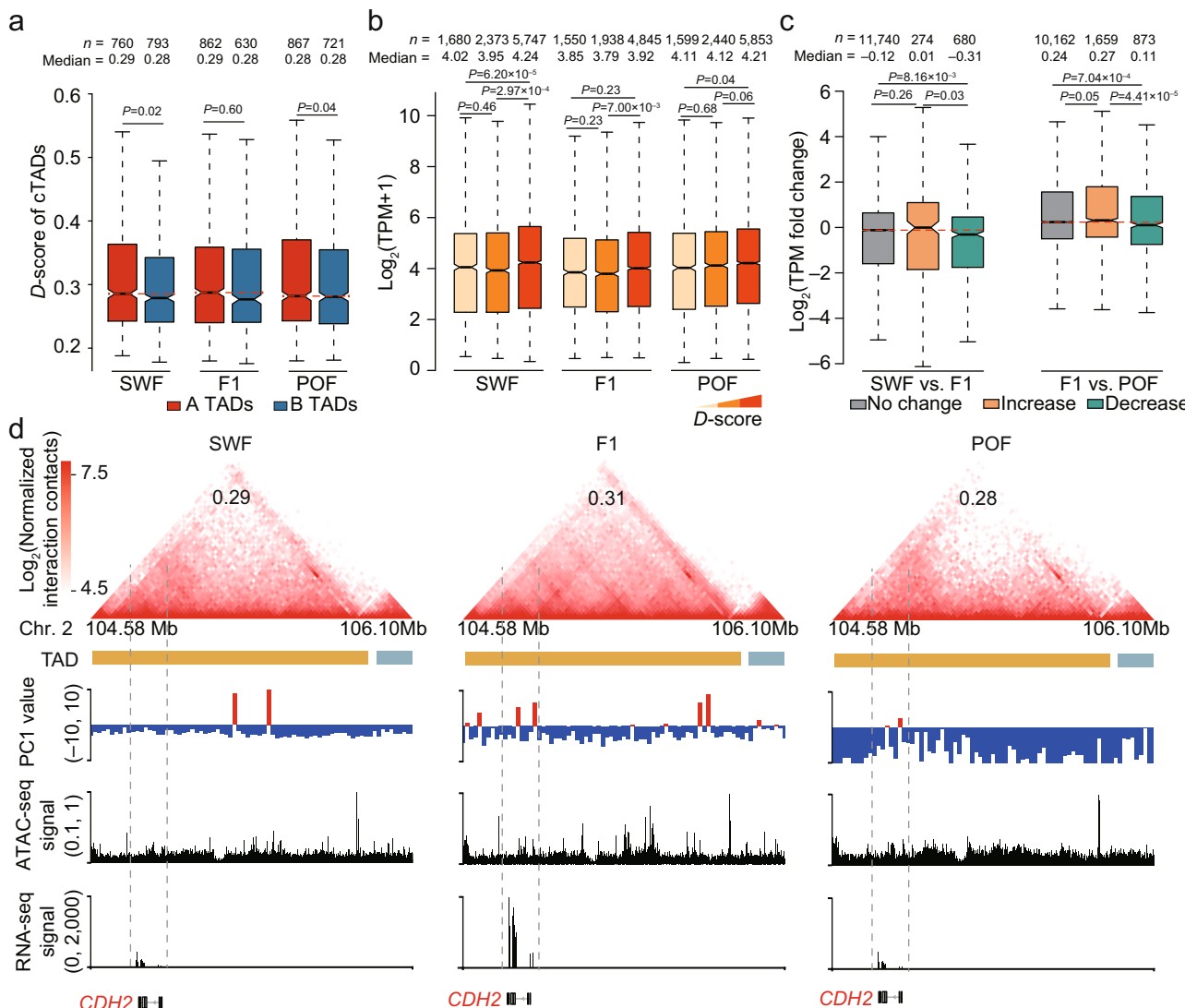

**Fig. 5 Interaction dynamics in consensus TADs during folliculogenesis in chicken GCs. a** *D*-scores of consensus TADs located in the A and B compartments at each stage. **b** Expression levels of genes in consensus TADs (*n* = 1831) with a relatively low, medium, or high *D*-score. **c** Expression changes in genes with significantly increased, decreased, or stable *D*-scores between adjacent stages. **d** A representative TAD with differential *D*-scores during folliculogenesis. Top: Hi-C contact heatmaps of the genomic region containing *CDH2* (Chr.2: 104.58–106.10 Mb). *D*-score values of the TADs were marked. Middle: TAD boundaries and genome browser tracks of PC1 values, ATAC-seq signals, and gene expression levels. Bottom: gene structures in the genomic region. The dashed line boxes indicate the chromosomal locations of the genes (red). For **a**, **b**, and **c**, the internal line indicates the median, the box limits indicate the upper and lower quartiles and the whiskers extend to 1.5 IQR from the quartiles. *P*-values in a were calculated using one-sided Student's *t* test. *P* values in **b** and **c** were calculated using two-sided Wilcoxon rank-sum test. Source data are provided as a Source Data file.

category. This generally reflected differences in the physiological features of follicles during folliculogenesis (Fig. 6d). Notably, 905 genes (666 and 239 genes in clusters 1 and 2, respectively) experienced increases in RPS at the SWF stage, which primarily involve developmental processes such as cell part morphogenesis. This reflects the activation of primordial follicles to primary follicles, which is accompanied by a change in GCs morphology (i.e., from flat, elongated to cuboidal shape) at the SWF stage[57]. A total of 1150 genes (224 and 926 genes in clusters 3 and 4, respectively) showed increases in RPS during the F1 stage, which primarily involve biological regulation and growth-related processes such as the regulation of hormone levels. Additionally, the RPSs of 604 genes (534 and 70 genes in clusters 5 and 6, respectively) were the highest in the POF stage, which primarily involve ubiquitin-dependent protein catabolic process, proteolysis involved in cellular protein catabolic processes, and autophagy.

This aligns with the structural breakdown and functional regression of GCs after ovulation in POF[58].

Our results demonstrated that candidate loci can be analyzed in future studies of folliculogenesis. Typically, *CCNA1*[59], *CDC20*[60], and *CPEB1*[61] (which are cell cycle regulators) and *BRDT*[62] and *FBXO43*[63] (which are involved in cell meiosis) showed more PEIs and contacted with SEs during the SWF stage, though these interactions and active enhancers decreased in later stages (Supplementary Fig. 12a–e). *PPARG*[64] and *SCAP*[65] (which are related to lipid homeostasis) and *FDX1*[66] and *SOCS2*[67] (which are associated with steroid hormone production) exhibited the most PEIs and were regulated by SEs during the F1 stage (Fig. 6e, Supplementary Fig. 12f–h). *FOS*[68] and *TIMP2*[69] (which are involved in the apoptotic process) interacted with SEs and displayed more PEIs at the POF stage than during other stages (Supplementary Fig. 12i, j). In addition, the convergent CTCF-

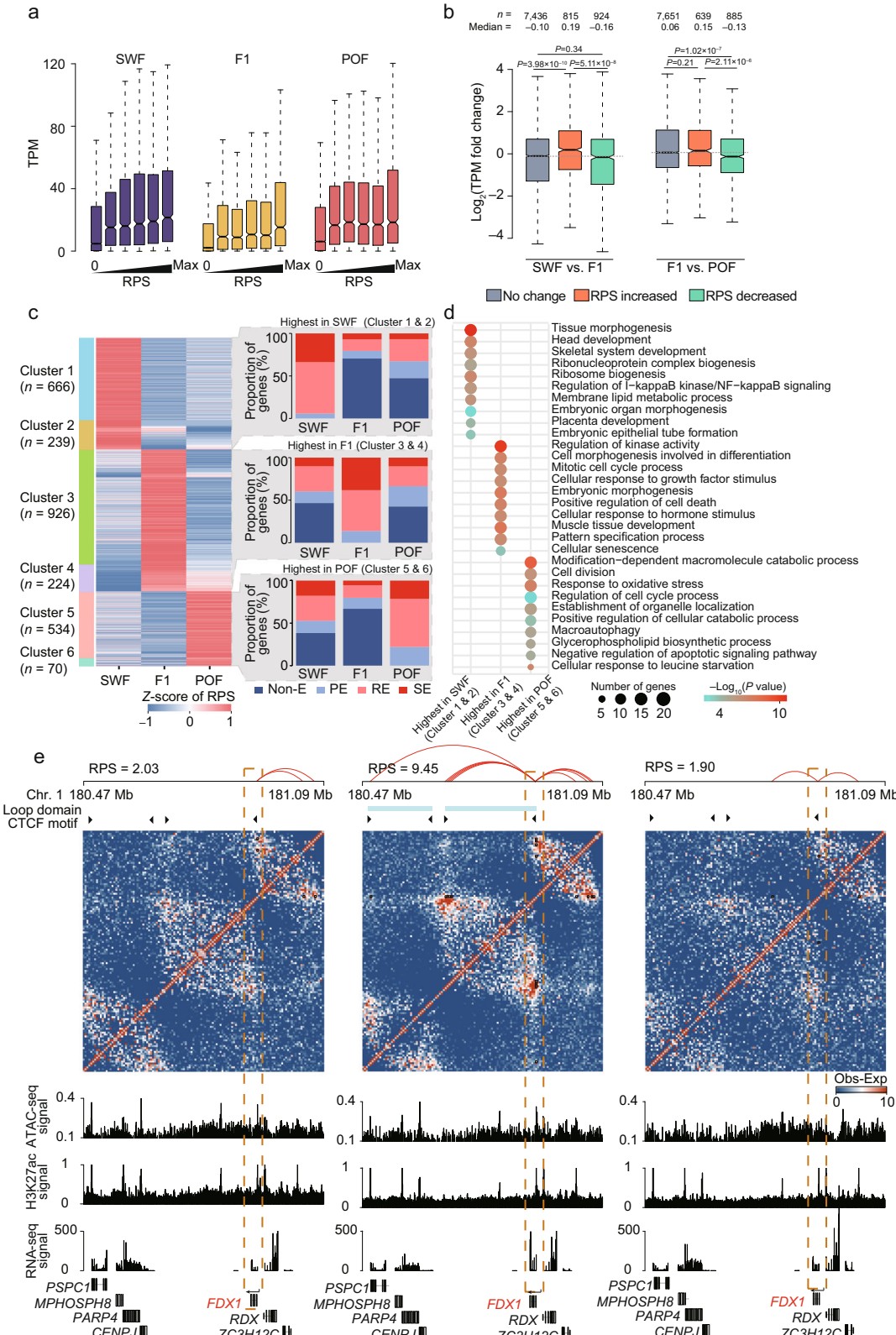

CTCF loop embedded with *FDX1* was only identified in the F1 stage, which is consistent with the most PEIs detected in this stage (Fig. 6e). This result indicates that CTCF sites are preferentially located near promoters and/or enhancers to constrain the PEIs (Supplementary Fig. 12k, l, Supplementary Data 5).

## Discussion

This study comprehensively analyzes Hi-C, RNA-seq, scRNA-seq, ChIP-seq, and ATAC-seq across a defined time course of chicken in vivo folliculogenesis. GO terms of stage-specific signature genes of GCs were identified by scRNA-seq and emphasized significant functional differences between the SWF, F1, and

**Fig. 6 Promoter-enhancer interactions (PEIs) rewired in chicken granulosa cells during folliculogenesis. a** Genes with higher regulatory potential scores (RPSs) show elevated expression levels. The genes are divided into six groups based on their RPS values (five quintiles of non-zero values and RPS equals zero). **b** Significant differences observed in expression levels of the genes with increased, decreased, or stable RPS values between adjacent stages. *P* values were calculated using two-sided Wilcoxon rank-sum test. **c** *K*-means clustering of the genes with RPS changes during folliculogenesis (*k* = 6). The proportions of genes interacting with super-enhancers (SEs), regular enhancers (REs), and poised enhancers (PEs) are displayed on the right. **d** The most enriched GO-BP terms for genes with high RPS at a specific stage. **e** PEI rewiring of a functional gene *FDX1* (red) during folliculogenesis. Top: schematics of PEIs and Hi-C contact heatmaps of the genomic region containing *FDX1* (Chr.1: 180.47–181.09 Mb). The light blue line indicates loop domains, while black arrowheads indicate CTCF motif orientation at loop anchors. Middle: genome browser tracks of ATAC-seq signals, H3K27ac signals, and gene expression levels. Bottom: gene structures in the region. The dashed line boxes indicate the chromosomal locations of the genes. For panels a and b, the internal line indicates the median, the box limits indicate the upper and lower quartiles and the whiskers extend to 1.5 IQR from the quartiles. Source data are provided as a Source Data file.

POF stages. These changes in gene expression, including *AMH*, reflected the corresponding functional characteristics of follicle development at different physiological stages and were consistent with other RNA-seq studies during chicken folliculogenesis[11–13,70]. The gradual establishment of the genome structure from the SWF to F1 stage likely affects the functional maturity of GCs, while the loose chromatin structure during the POF stage could initiate the apoptosis⁻related pathway[27].

Notably, we found that higher-order chromatin structures, including compartmentalization, TAD boundaries, and intra-TAD interactions, were dynamic during the stage transformation of GCs and were associated with gene expression changes. At the A/B compartment level, we observed moderate compartment switches and subtle changes in gene expression, which corresponded to the A/B switches around the development of the prehierarchical to hierarchical and preovulatory to postovulatory stages. These results suggest that dynamic chromatin compartmentalization and limited gene expression alteration (inferred from mRNA levels) occurs during folliculogenesis, including expression changes in key functional genes such as *AMH*[71] and *LHCGR*[72]. This finding agrees with the Hi-C study of stem cell differentiation[31] and indicates a relatively conserved mechanism for gene regulation mediated by dynamic compartmentalization.

At the TAD level, the number of TAD boundaries we obtained in chicken GCs (~2000) is comparable to those identified in chicken fibroblasts (~2000) by the DI algorithm[17]. These observations differed from the chromatin architectures of chicken erythrocytes (~1300), which display neither a typical TAD structure nor long-range chromatin interactions. This suggests the specificity of this cell type[17], even though single-cell Hi-C experiments have demonstrated that TAD is a genuine unit of chromatin folding[73]. It has been reported that TADs largely remain stable across tissues in many organisms[44,74]. Indeed, we found that only a few TAD boundaries dynamically changed during folliculogenesis. Although most TAD boundaries were unchanged, we did observe a subset of TADs with dynamic interaction frequencies during folliculogenesis and found that these *D*-score changes were accompanied by dynamic gene transcriptional activities. Our results support current findings that the synergistic effect and boundary insulation of TADs could be an important mechanism underlying regulation of gene expression and that TADs might limit the physical interactions of transcriptional regulatory elements such as enhancers and promoters to construct autonomous gene regulatory regions[75].

At the PEI level, we identified long-range PEIs with ultra-deep Hi-C contact maps to accurately investigate the gene transcriptional regulation and cellular functions of GCs. Similar to the findings in mammals[53,76,77], enhancers can skip the proximal promoters and interact with distal genes, highlighting the complexity of the regulatory landscape in chickens. Stage-increased enrichment of RPS demonstrated that the remodeling of PEIs in GCs modulates gene transcription during folliculogenesis.

In conclusion, these results indicate that transcriptomic and chromatin architectural changes in GCs during folliculogenesis could facilitate the concomitant transcriptional activities and provide a valuable resource that allows the in-depth functional characterization of GCs. We primarily assessed GCs in this study. Advances in sequencing technology at single-cell resolution imply that further research is needed to explore the 3D genome architectural dynamics and how it influences gene expression in oocytes throughout the reproductive cycle, as well as analyze various epigenetic data to better understand regulatory mechanisms in follicle development.

## Methods

**Ethics statement**. All animal protocols were approved by the Institutional Animal Care and Use Committee of Sichuan Agricultural University (protocol number B20171910). The methods were carried out in accordance with the approved guidelines.

**Animals and sample collection**. Healthy Luhua hens at peak egg-laying age (31 weeks of age) were euthanized via the intravenous injection of 2% pentobarbital sodium (25 mg/kg of body weight). Follicular GCs layers of the whole reproductive cycle, including prehierarchical follicles (small white follicle: SWF; large white follicle: LWF; small yellow follicle: SYF; large yellow follicle: LYF), preovulatory follicles (F5, F4, F3, F2, and F1), and postovulatory follicles (POFs), were collected as described by Gilbert. et al.[10]. In detail, once the ovaries were harvested, the ovarian follicles were carefully excised. The stalk of the excised follicle was held with forceps so that the clear, avascular stigma was on top and a cut was made with a scalpel approximately along the line of the stigma quickly. Immediately after it was cut, the follicle was inverted over phosphate buffer solution (PBS) and the follicles were gently shaken to remove the yolk. The follicles were gently shaken until a transparent film appeared in the PBS, which is the GC layers. All separated GCs samples were promptly frozen in liquid nitrogen and stored at −80 °C for subsequent assays (i.e., bulk RNA-seq, scRNA-seq, in situ Hi-C, ATAC-seq, and ChIP-seq).

**Bulk RNA-seq**. Total RNA was extracted from each sample using RNAiso Plus reagent (TaKaRa, #9108) according to the manufacturer's instructions. We estimated the integrity and quality of the total RNA using a Bioanalyzer 2100 system (Agilent Technologies, Palo Alto, CA, USA) and an RNA 6000 Nano kit. Sixty poly-A RNA-seq libraries were constructed. They were then sequenced using the BGISEQ DNBSEQ-T7 platform (BGI lnc., Shenzhen, China) with a paired-end sequencing length of 150 bp (PE150) at Novogene Bioinformatics Technology Co., Ltd (Beijing, China).

**Gene expression analyses**. The high-quality reads were mapped to the chicken reference genome (GRCg6a) using HISAT2 2.1.0[78], and the uniquely mapped reads were assembled and quantified using StringTie v1.3.3 to assess gene expression based on the TPM (Transcripts Per Million) values of each mRNA[79]. Spearman correlations were calculated across the developmental stages. The DEGs were identified using DEseq2[80] based on the read count data. The significant DEGs were screened with a false discovery rate <0.05 and |log₂fold change| > 1 as cutoffs. MaSigPro (v 3.12)[19] was used to identify genes with dynamic temporal expression profiles.

**Cell preparation of scRNA-seq**. The separated GC layers were washed in ice-cold RPMI1640 and dissociated using a multi-tissue dissociation kit 2 (Miltenyi, #130-110-203) from Miltenyi Biotec, according to the manufacturer's instructions. DNase treatment was optional depending on the viscosity of the homogenate. Cell count and viability were estimated using fluorescence Cell Analyzer (Countstar®

Rigel S2) with AO/PI reagent after the erythrocytes were removed (Miltenyi, #130-094-183), while the debris and dead cells were removed (Miltenyi, #130-109-398). Finally, fresh cells were washed twice in the RPMI1640 solution and resuspended at $1 \times 10^6$ cells per ml in $1 \times$ PBS and 0.04% bovine serum albuminate.

**scRNA-seq library construction and sequencing**. scRNA-seq libraries were prepared according to the manufacturer's instructions of Beijing SeekGene BioSciences Co., Ltd (Beijing, China). Libraries were prepared using Chromium Next GEM Single Cell 3′ Reagent Kits v3.1 (10× Genomics). The appropriate number of cells were mixed with reverse transcription reagents and then loaded to the sample well in a Chromium Next GEM Chip G. Gel Beads and Partitioning Oil were then dispensed into corresponding wells, separately in a chip. We then performed emulsion droplet generation reverse transcription at 53 °C for 45 min and inactivated at 85 °C for 5 min. The cDNA was then purified from the broken droplet and amplified in a PCR reaction. The amplified cDNA products were then cleaned, fragmented, end-repaired, A-tailed, and ligated to the sequencing adapter. Finally, the indexed PCR was performed to amplify the DNA representing 3′ polyA part of expressing genes which also contained Cell Bar code and Unique Molecular Index. The indexed sequencing libraries were cleaned with SPRI beads, quantified by quantitative PCR (KAPA Biosystems KK4824), and sequenced on an Illumina NovaSeq 6000 with PE150 read length.

**scRNA-seq data processing and analysis**. Raw sequencing data were processed using Cell Ranger analysis pipeline (v2.1.1). Reads were aligned to the chicken genome version GRCg6a. For downstream analysis, we used the Cell Ranger output "filtered gene-barcode" count matrix, which contained the expression profile of cells with a properly detected cellular barcode. To explore the purity and markers of GCs, we used the available high-quality scRNA-seq data of chicken heart cells as comparative controls[20] for analysis. We further used the R package Seurat, v2.2.014[81] with the following parameters to filter high-quality cells and exclude cells with extreme values indicating low complexity, duplets, or apoptotic cells: the total number of expressed genes/cell was 1200 < nGenes < 5000; the total number of UMIs/cell was nUMIs > 3000, and the percentage of UMIs mapped to mitochondrial genes to total genes was <0.2. Counts were normalized using Seurat (Function NormalizeData) at default settings. For each cell, the UMI counts for each gene were divided by the sum of the UMI counts for all genes for that cell. The results were multiplied by a fixed factor (10,000) and $\log_e$-transformed. For paired-wise differential expression analysis, Function FindAllMarkers from Seurat was performed between the cells of a cluster and the rest of the cells in the dataset. The list of DEGs per cluster was identified with false discovery rate <0.05 (Wilcoxon rank-sum test) and |fold change| > 2. We used Function SplitDotPlotGG from Seurat to generate the dot plot. For each cluster, the mean expression of all genes was calculated and the fifty most variable gene means were selected using R function rowVars (package genefilter 1.60.050).

**In situ Hi-C**. In situ Hi-C library preparation was performed according to previously described methods[82] by Novogene Bioinformatics Technology Co., Ltd. Briefly, the follicular GC layers from different stages were homogenized with liquid nitrogen and then cross-linked with 4% formaldehyde for 30 min at room temperature (25 °C). The fixation reaction was quenched using 0.25 mol/L glycine for 5 min at room temperature after which it was placed on ice for 15 min. The nuclei of formaldehyde-fixed GCs were permeabilized, while the DNA was digested with 200 units of MboI (a 4-cutter restriction enzyme) for 1.5 h at 37 °C. The restriction fragment overhangs were filled and labeled with biotinylated nucleotides and then ligated in a small volume. Following cross-link reversal, the ligated DNA was purified and sheared to a length of 300–500 bp, after which the point ligation junctions were pulled down with streptavidin beads and prepped for the Illumina sequencing platform. Each library was sequenced on a Illumina HiSeq X Ten system with 150 bp paired-end sequencing read lengths.

**Hi-C data processing**. Hi-C datasets were analyzed using a custom Juicer pipeline (v 1.5)[83]. Briefly, the high-quality Hi-C reads were mapped to the chicken genome (GRCg6a) using BWA (v 0.7.15). Aligned read pairs were distributed to restriction motif fragments. After filtering duplicates, low-quality alignment read pairs (MAPQ < 30), and intrafragment read pairs, we obtained the valid Hi-C read pairs. We then generated contact matrices at 20 kb and 5 kb resolution and normalized the matrix using the KR algorithm and used HiCRep[84], GenomeDISCO, and QuASR-Rep[85] to assess the reproducibility of the Hi-C data.

**Identification of A/B compartments**. We identified the A and B compartments using a 20 kb resolution interaction matrix as previously described[14]. In detail, principal component analysis was performed to generate the PC1 vectors of the chromosomes from each sample, and the Spearman correlations between PC1 and genomic characteristics (gene density and GC content) were calculated. Bins with a positive Spearman's $r$ were defined as compartment A, while remaining bins were defined as compartment B.

**Chromatin 3D modeling**. The 3D chromosome conformations were inferred for each Hi-C map based on the normalized intra- (at 100 kb resolution) and inter-chromosomal (at 1 Mb resolution) contact maps using an approximation of multidimensional scaling (MDS) method as implemented in miniMDS[28] program. After modeling, through Euclidean distance to measure the relative distance of each chromosome (100 kb resolution) to nucleolus (start point).

**Identification of topologically associated domains (TADs)**. TADs were identified with 20 kb resolution matrices using the DI[44] and IS[45]. First, DI was calculated for 10 bins upstream and 10 bins downstream from the center of each bin, at 20 kb resolution. The hidden Markov model (HMM) was then used to predict the DI states for the TAD border. Then, based on the TAD identified by DI, we used the minimal IS (along the normalized insulation score vector) to further divide the large TADs into small TADs. To determine the A/B status of a TAD, we counted the frequency of the A/B bin membership (PC1) within the TAD and defined the TAD as an A-TAD if more than 70% of the TAD belonged to the A compartment and defined it as a B-TAD if not.

**Analysis of TAD**. Specific TAD boundaries for each stage were identified as previously reported[44]. To compare the TAD boundaries of two stages, we first merged the center positions of the TAD boundaries of the two stages and calculated the DI 10 bins upstream and 10 bins downstream (±200 kb) from the center of each boundary. Spearman's correlation coefficient was calculated from 20 randomly selected bins between each adjacent stage for the random correlation, while the randomization was repeated 1,000 times to obtain the random distribution of Spearman correlation coefficients. A specific boundary is defined as a boundary identified at only one stage that does not significantly differ from a random correlation distribution. Genes near specific TAD boundaries (such as genes within the boundary and flanking two bins) were affected by TAD location movement.

To quantify the interaction strength within TADs by domain score ($D$-score)[49], we analyzed the consensus TADs (cTADs) in GCs across different stages, which were conserved in at least 50% of all developmental time points and replicates. We then defined the $D$-score of a consensus TAD as the ratio of intra-TAD interactions over all intrachromosomal interactions for the consensus TAD. To identify TADs with differential interaction strengths between adjacent stages, we performed Student's $t$ tests to compare $D$-scores, while TADs with a $P$ value less than 0.05 were selected for further analysis.

**Identification of promoter-enhancer interaction (PEI)**. To identify the PEIs of each gene, we merged the Hi-C reads from the biological replicates of each stage and generated 5 kb resolution contact matrices. The normalized contact matrix was split into a smaller matrix (20 Mb × 20 Mb) with 10 Mb steps of overlapping length, which were subsequently analyzed with PSYCHIC[51] at default parameters to identify overrepresented interactions with the promoter region. We reserved highly confident PEIs with FDR values <0.01 and interaction distances ≥15 kb. To characterize the additive effect of enhancers on gene regulation, we calculated the regulatory potential score (RPS) for each gene, which is defined as follows: the sum of all significant interaction intensities ($\log_{10}$(observed number of contacts − expected number of contacts)). We defined differential RPS using the cutoff |$\log_2$FC| > 1.5 and |ΔRPS| > 3. The genes with differential RPS values were clustered by $K$-means.

**Convergent CTCF-CTCF loops identification**. We used the FIMO software (v5.1.1)[86] to identify CTCF motif loci and their orientations in the GRCg6a version of the chicken genome based on consensus CTCF motif from the JASPAR CORE 2016 vertebrate database[87]. As expected, these CTCFs were enriched in TAD boundaries and enhancer regions. We also separately identified loops in SWF, F1, and POF at 5 kb resolution in the genomic distance range of 20 kb to 2 Mb using the Fit-Hi-C Python package (v2.0.7) ($q$ value < 0.05)[88]. To further obtain the highly confident loops, we applied a hard cutoff to obtain the top 15,000 loops by ranking the loop strengths. We finally obtained 6632, 7050, and 8023 convergent CTCF-CTCF loops in SWF, F1, and POF, respectively. The genes in these loops have 4067, 4926, and 2582 PEIs, respectively.

**ATAC-seq**. We used an improved ATAC-seq protocol compatible with cryopreserved samples[89,90]. Cryopreserved samples were washed with 0.09% NaCl solution and ground into powders in liquid nitrogen. A lysis buffer was added to the powders and incubated on ice for 10 min on the rotation mixer. The cell suspension was then filtered with a 40 μm cell strainer and washed with cold PBS buffer three times to collect the cell nuclei. Approximate 50,000 nuclei were allocated to perform tagmentation according to standard protocols[89], while the Tn5 transposed DNA was purified by AMPure DNA magnetic beads. Next, an average of 11 cycles of PCR was performed with transposed DNA, and amplified libraries were run on an Agilent TapeStation 2200 (Agilent Technologies) using a D5000 DNA ScreenTape to assess their quality via visualizing nucleosomal laddering. The final libraries were submitted for sequencing on the Illumina HiSeq X Ten platform with 150 PE modes.

**ATAC-seq data analysis**. Quality control of raw sequencing data was performed using Cutadapt (v 1.9.1). Trimmed reads were aligned to the chicken genome (GRCg6a) using Bowtie2 (v 2.2.6) with default settings[91]. Mitochondrial alignments were removed using removeChrom (https://github.com/jsh58/harvard), and Picard (v 1.126) (http://broadinstitute.github.io/picard/) was used to remove PCR duplicates and for insert size distribution analysis. Finally, broad peaks were called using MACS2 with options "–nomodel –extsize 200 –shift –100"[92]. We also checked the enrichment of ATAC peaks on transcription start site[93] regions and confirmed the correlation between ATAC peaks and A/B compartments. For stage-specific peaks, we merged all called peaks from all samples to acquire the consensus peak-set. We defined the enrichments as ATAC signal intensity (normalized read count per base) subtracted by the background noise (normalized read count per base). The R package DESeq2 was used to detect potential stage-specific peaks. Peaks with $|\log_2 \text{fold change}| > 1$ and $\text{FDR} < 0.05$ were considered significantly different. Motif enrichment analysis was performed using MEME suite[94] with default settings. Motifs with $P$ value $< 0.01$ were kept.

**ChIP-seq**. We performed ChIP-seq using an antibody against H3K27ac for two replicates in each of the three stages (SWF, F1, and POF). GCs from the ovary were washed twice in cold PBS buffer, cross-linked with 1% formaldehyde for 10 min at room temperature, and then quenched by adding glycine (125 mmol/L final concentration). Afterward, the samples were lysed and chromatins were kept on ice. Chromatins were sonicated to obtain soluble sheared chromatin (average DNA length of 200–500 bp), after which 20 μL chromatin was saved at −20 °C for input DNA and 100 μL chromatin was used for immunoprecipitation by Rabbit polyclonal anti-Histone H3K27ac (Abcam, #ab4729). A total of 5 μg of antibodies were used in immunoprecipitation reactions at 4 °C overnight. The next day, 30 μL of protein beads were added and the samples were further incubated for 3 h. The beads were washed once with 20 mM Tris/HCL (pH 8.1), 50 mM NaCl, 2 mM EDTA, 1% Triton X-100, 0.1% SDS; twice with 10 mM Tris/HCL (pH 8.1), 250 mM LiCl, 1 mM EDTA, 1% NP-40, 1% deoxycholic acid; and twice with 1×TE buffer. Bound material was then eluted from the beads in 300 μL of elution buffer (100 mM NaHCO₃, 1% SDS), treated with RNase A (final concentration 8 μg/mL) for 6 h at 65 °C, and then treated with proteinase K (final concentration 345 μg/mL) overnight at 45 °C. Immunoprecipitated DNA was used to construct sequencing libraries according to the instructions provided by the I NEXTFLEX® ChIP-Seq Library Prep Kit for Illumina® Sequencing (NOVA-5143, Bioo Scientific) and sequenced on an Illumina HiSeq X Ten platform with 150 PE modes.

**ChIP-seq data processing**. Trimmomatic (v 0.38) was used to filter out low-quality reads[95]. High-quality reads were mapped to the GRCg6a version of the chicken reference genome by BWA (v 0.7.15), allowing up to two mismatches. Samtools (v 1.3.1) was used to remove potential PCR duplicates, and the MACS2 algorithm (http://liulab.dfci.harvard.edu/MACS/) was used to call H3K27ac peaks by default parameters (bandwidth, 300 bp; model fold, 5, 50; q value, 0.05). We defined super-enhancers (SEs) using the standard ROSE algorithms. Briefly, neighboring enhancer elements (within 12.5 kb) were defined based on H3K27ac and ChIP-seq peaks were merged and ranked by H3K27ac signals to identify an inflection point. Enhancers above the inflection point were defined as SE peaks, and those below the inflection point were defined as regular-enhancer (RE) peaks. Genomic regions contacting distal promoters but not overlapping with H3K27ac peaks were defined as PEs.

**Functional enrichment analysis**. Functional enrichment analyses were performed using Metascape (http://metascape.org)[96] with default parameters. Chicken genes were converted to human orthologs, and the target gene lists were uploaded as inputs for enrichment. We chose humans (*Homo sapiens*) as the target species, and enrichment analysis was performed against all genes in the genome as the background set, with the biological process (BP) of Gene Ontology (GO) as the functional test set. Only GO terms with a $P$ value $< 0.01$ and annotated to ≥3 genes were considered significant.

**Statistical analyses**. All statistical analyses were performed by Student's $t$ test, Wilcoxon rank-sum test or Mann-Whitney $U$ test using R.

**Reporting summary**. Further information on research design is available in the Nature Research Reporting Summary linked to this article.

## Data availability

The Hi-C data of granulosa cells (GCs) generated in this study have been deposited in the GEO database under accession code "GSE167064". The bulk RNA-seq, single-cell RNA-seq, ATAC-seq, and ChIP-seq data generated in this study have been deposited in the GEO database under accession code "GSE181756". Chicken hearts single-cell RNA-seq data was downloaded from GEO database under accession codes "GSE149457". Chicken embryonic fibroblasts, chicken immature and mature erythrocytes Hi-C data was downloaded from GEO database under accession codes "GSE96037". All other data supporting the findings of this study are available within the article and its Supplementary Information files or from the corresponding author upon reasonable

request. A reporting summary for this Article is available as a Supplementary Information file. Source data are provided with this paper.

## Code availability

Each use of software programs has been clearly indicated and information on the options that were used is provided in the Methods. All software, codes, and scripts used for data processing and analyses are available on GitHub through the following link, https://github.com/JiamanZhang/Lab_GCs_paper_codes, or at https://doi.org/10.5281/zenodo.5677410.

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

## Acknowledgements

We appreciate the High-Performance Computing Platform of Sichuan Agricultural University and the Ya'an Big Data Industrial Park for providing computing resources and support that have contributed to these research results. This work was supported by the National Key R & D Program of China Grant 2020YFA0509500 (to M.L.), the National Natural Science Foundation of China Grants 31972543 (to Q.Z.), U19A2036, 31872335, and 31772576 (to M.L.), the Sichuan Science and Technology Program Grants 2019JDTD0009 (to D.L.), 2020YFSY0040 (to T.W.) and 2021YFYZ0009 (to M.L.), the Ya'an Science and Technology Program Grant 21SXHZ0022 (to L.J.).

## Author contributions

D.L. and M.L. conceived the study and the analytical strategy. C.N., M.H., Y.Z., X.Z., H.X., and H.K. collected biological samples. J.Z., Y.L., B.Z., Q.T., L.J., H.Y., and T.W. designed the bioinformatics analysis process. J.Z. and C.N. performed data analysis. D.L., C.N., J.Z., and Y.W. wrote the paper. D.L., M.L., Q.Z., and J.L. revised the paper.

## Competing interests

The authors declare no competing interests.

## Additional information

**Peer review information** *Nature Communications* thanks Tatjana Sauka-Spengler and the other anonymous reviewer(s) for their contribution to the peer review this work. Peer reviewer reports are available.

