## [Peer Review File · Nature Communications]

Reviewers' Comments:

Reviewer #2:

Remarks to the Author:

In the manuscript "Dynamic chromatin architecture in ovarian follicle development in chickens" authors provided a detailed analysis of gene expression profiles and chromatin interaction maps during chicken follicle development. Chicken growing follicles represent a favorable model for gene expression analysis. Investigation of gene transcription regulation at the level of genome 3D-organization in chicken follicles could help to improve laying performance.

Gene expression analysis in 10 ovarian follicle developmental stages allowed authors to classify genes according to their stage specific expression and functional role. At 3 selected developmental stages RNAseq data was correlated with HiC chromatin contact maps. Authors found switches between compartment types during follicle development but did not find strong correlation with gene expression pattern. Structural stability of identified TADs was also demonstrated as expected. Most interesting findings are related to identification of changes in interactions between distant regulatory elements during follicle development.

Authors represent high quality original data that is clearly summarized in 6 main and 10 supplementary figures. The manuscript is logically written and the data for the most part are convincing. However the following questions should be addressed:

Questions:

Q1. Hi-C is very sensitive to cells purity. How authors controlled purity of the cell populations used in this study, and what is possible percentage of cross-contamination between them?

Q2. "Focusing on the TAD boundaries across the development of follicles, the intensified spatial segregation at the F1 stage indicated that more self-interactions occurred within TADs when the follicle reached maturity (Fig. 4d-f)." – the portion of intra-TAD interactions may reflect noise level and/or P(s) scaling, which is slightly different between samples. For example, it is expectable that higher noise level will lead to higher portion of intra-TAD interactions. This possibility should be excluded before drawing biological conclusions from very slight differences shown in Fig. 4 d and e. For example, note that in Fig. 4,E at the point -300 kb, the difference in insulation between POF and other stages is higher than in the TAD boundary region.

Q3. Promoter-enhancer interactions should be distinguished from CTCF-based loops formed via loop extrusion mechanism. It is known that CTCF sites are preferentially located near promoters and/or enhancers. Consistent with this, it seems that most of interactions depicted in Fig. 6, G and called by PSYCHIC connect TAD boundaries. These CTCF-mediated interactions might have regulatory effect; however, acute degradation of CTCF protein (Rao et al., 2017) showed that there are "bona fide" PEIs, which are mediated by mechanisms other than CTCF blocked extrusion. Authors should carefully distinguish these two types of interactions through the whole paragraph "Global remodeling of promoter-enhancer interactions in transcription control during follicle development".

Q4. "The spatial organization of compartments constructed by miniMDS35 showed that PC1 value and gene expression are negatively associated with the distance from the center of the nucleus, i.e., the nuclear radius" – it is not clear how the nuclear radius was measured in each cell type.

Major comments:

C1. The Introduction could be improved and better structured. In the current version Introduction ends with the too long section summarizing main results. By my opinion, major goal of the study is also not clearly stated in the Introduction.

C2. Please compare the RNAseq data obtained for chicken ovarian follicles with previously published studies describing chicken follicle transcriptome.

C3. Please compare HiC data and identified TAD borders in chicken granulosa cells with other chicken cell types.

Minor comments:

C1. By my opinion, the phrase «dramatic remodeling of genome structures» is too strong for the data obtained.

C2. Within the Abstract: "we integrated RNA-seq and Hi-C analyses of chicken follicular granulosa cells of 10 developmental stages." please correct the sentence so that it would be clear that HiC analyses was performed in 3 developmental stages.

C3. Fig 1b. Indicate in the figure legend what is shown by lines of different color.
C4. Fig 4h. Please show the HiC-contact heatmaps without TAD borders.

Reviewer #3:

Remarks to the Author:

The manuscript "Dynamic chromatin architecture in ovarian follicle development in Chickens " is an interesting and well written report. The authors integrated RNA-seq and Hi-C analyses of chicken follicular granulosa cells of 10 developmental stages. Overall, the authors identified novel mechanisms in chickens, such as spatial organization of compartments in the nucleus and correlation with gene expression, similar to what has been observed in mammals. The data is presented clearly and follows a logical path.

One of the main concerns with this report is the analysis of the follicles as a whole unit, without the differentiation between the somatic and the germ compartments.

Selecting small white follicles (SWF), preovulatory (F1) and postovulatory follicles (POF) for the generation of genome-wide contact maps of chicken follicle development using Hi-C technology raises the question of how the authors compare SWF and F1, that contain an oocyte, with POFs that are devoid of oocytes. What is the contribution of the oocyte to the reported data?

In small follicles, I would assume that the oocyte contributes to 50% of the follicle. How authors separate the RNA contribution from the somatic versus germ cells to their data set?

Oocytes undergo significant changes during folliculogenesis, but the authors only reported data regarding granulosa cells without in-depth discussion of what is missing without the separate analysis of the oocytes.

The authors did not mention any limitations of the study. One limitation I could think about is the lack of the germ cell analysis.

Abstract – please avoid using acronyms, such as SWF, F1, POF and TAD, in the abstract .

Reviewer #4:

Remarks to the Author:

This work profiles the transcriptional dynamics of chicken follicular granulosa cells at ten developmental stages and chromatin folding dynamics using Hi-C at three critical stages (Small white follicle (SWF), growing preovulatory follicle stage F1 and postovulatory follicles (POF) stage SWF to POF using RNA-seq and characterised three stages (SWF, F1 and POF). Subsequently, the authors perform analyses including chromatin compartmentalisation, TADs and promoter-enhancer interactions and make a valid attempt towards integrating transcriptional and Hi-C data. Finally, the authors present two examples of local chromatin remodelling leading to expression upregulation during this developmental process. Albeit descriptive, this study provides global profiling of chromatin structures in chicken ovarian follicle development and represents a worthy contribution to the field. The literature lacks comprehensive transcriptomics and chromatin folding analysis across the different developmental stages of the chicken ovarian follicles. This data will be beneficial for further future studies of gene regulatory processes in the ovarian follicles.

Major concerns:

1. Line 312-317. The authors point out a strong correlation between changes in chromatin compartmentalisation but remark that the corresponding shifts in gene expression levels are missing. The main concern here is that this results from the coarse-grained resolution of chromatin compartmentalisation analysis currently presented in the paper. Ideally, authors need to increase the resolution by much deeper sequencing or perform targeted chromosome conformation capture experiments for several specific loci (4-C or Capture-C, for example) to address this issue. Furthermore, the authors should address this point in the discussion. The authors should comprehensively discuss to which extent interactions at various resolutions (e.g., A/B

compartment, TAD and PEI) respectively contribute to changes in gene expression.

2. RNA-seq readout corresponds to mRNA levels. Both transcriptional activities and RNA degradation affect mRNA levels. The authors should discuss the relationships between changes in chromatin compartmentalisation and alterations in mRNA levels.

3. The authors should add a two-dimensional scatter plot to visualise the global relationships among RNA-seq samples

4. The authors have used maSigPro to cluster all genes to globally characterise transcriptomic changes and then used GO analysis to summarise the functions of genes assigned to each cluster. A well-known limitation of GO analysis is its over-generalisation and vagueness. Such an example is, for instance, the enrichment in the "growth and developmental processes" GO term. The authors should include a heatmap of enriched/representative genes that define particular terms between stages to replace fig 1d. Another option would be to use bubble plots to include both GO term enrichment and statistical significance values in the exact representation.

5. Given the study's descriptive nature and lack of resolution in chromatin data, the authors should define specific active putative cis-regulatory elements by adding chromatin accessibility (ATAC-seq) characterisation at selected time points (same as Hi-C analysis). Integration of these data would greatly help understand the intricacies of the gene regulatory programme underlying chicken ovarian follicle development.

6. It is surprising that using pairwise differential expression analysis, the authors found "no difference in gene expression between contiguous stages of development" (line 139-140), especially between SWF/LWF, LYF/F5 and F1/POF. Increasing the sequencing depth would likely for RNA-seq libraries as well allow identification of the differentially expressed genes between these stages (see comment above)

7. As admitted by the authors, "chromatin compartmentalisation contributes to relatively subtle changes in gene expression and does not play a deterministic role" (line 314-317). Hence, by definition, the significance of this work in terms of understanding the determinants of gene expression during follicle development is limited. One way to possibly improve the study's depth, bringing new insights, would be by extending the transcriptomic analysis using single-cell RNAseq on selected developmental stages - the study would then provide an atlas of both transcriptome and genome-wide chromatin interactions during chicken ovarian follicle development.

Minor concerns:

1. Fig 1b legend. It is not clear what the lines denote.

2. Fig 1c. It is a very informative figure validating the temporal expression of representative genes from each cluster. First, the clusters can be ordered by cluster 2, 1, 4 and 3 to visualise the transition more clearly. Second, the vital genes of each cluster can be noted on the side of this heatmap.

3. Fig 2e. Authors should state the ratio of medians in addition to the p-value, as the low p-value can be mainly caused by a large number of sample sizes.

4. The details of multiple testing correction and other statistical methods should be clearly noted in the text, methods and figure legends.

5. Page 16 Line 310 typo: "an transient transition."

Detailed responses to reviewers

All comments provided by reviewers are given in gray italics, and our responses are in black. All revisions in the manuscript are marked in red.

Reviewer #1:

In the manuscript “Dynamic chromatin architecture in ovarian follicle development in chickens” authors provided a detailed analysis of gene expression profiles and chromatin interaction maps during chicken follicle development. Chicken growing follicles represent a favorable model for gene expression analysis. Investigation of gene transcription regulation at the level of genome 3D-organization in chicken follicles could help to improve laying performance.

Gene expression analysis in 10 ovarian follicle developmental stages allowed authors to classify genes according to their stage specific expression and functional role. At 3 selected developmental stages RNAseq data was correlated with HiC chromatin contact maps. Authors found switches between compartment types during follicle development but did not find strong correlation with gene expression pattern. Structural stability of identified TADs was also demonstrated as expected. Most interesting findings are related to identification of changes in interactions between distant regulatory elements during follicle development.

Authors represent high quality original data that is clearly summarized in 6 main and 10 supplementary figures. The manuscript is logically written and the data for the most part are convincing. However, the following questions should be addressed:

Comment 1-1:

Hi-C is very sensitive to cells purity. How authors controlled purity of the cell populations used in this study, and what is possible percentage of cross-

contamination between them?

Response 1-1:

Thank you for this comment.

We took steps to control the purity of granulosa cells (GCs). First, the follicular GCs were collected according to the methods described by Gilbert. et al. (Gilbert et al., 1977). We have added the below descriptions to the Methods section of the main text (Main text page 39, lines 650-661). The follicles were carefully excised from the ovaries of birds under general anesthesia or euthanized with sodium pentobarbitone. The stalk of the excised follicle was held with a forceps so that the clear, avascular stigma was on top and a cut was made with a scalpel approximately along the line of the stigma quickly. Immediately after it was cut, the follicle was inverted over phosphate buffer solution (PBS) and the follicles were shaken to remove the yolk. The follicles were gently shaken until a transparent film appeared in the PBS, which is the granular cell.

Furthermore, to demonstrate the purity of the GC samples, we constructed single-cell libraries for GCs at the SWF, F1, and POF stages. After quality control and filtering, a total of 21,393 high-quality cells (6,596, 5,996, and 8,801 cells for SWF, F1, and POF stages, respectively) were available for cell-type characterization. By integrating scRNA-seq data of developing chicken hearts with our data, we classified the cells into 13 clusters using the unbiased clustering and uniform manifold approximation and projection (UMAP) methods, generated hierarchical clustering using the 50 most variably expressed gene means per cluster, and finally distinguished two major cell groups: the GC group (three clusters) and other cells (ten clusters). To confirm the identity of these GCs, we colored the single cells according to the expression levels of five canonical markers of GCs (*CYP11A1*, *CHST8*, *FSHR*, *TSPAN6*, and *DSP*) and five representative genes (*NOV*, *RLN3*, *EDN2*, *FGL2*, and *RGS16*) specifically high expressed in GCs. We found that almost all (>99%) sequenced cells at the three stages were annotated as GCs. These results suggested that the isolated GCs had little contamination and high purity.

Comment 1-2:

“Focusing on the TAD boundaries across the development of follicles, the intensified spatial segregation at the F1 stage indicated that more self-interactions occurred within TADs when the follicle reached maturity (Fig. 4d-f).” – the portion of intra-TAD interactions may reflect noise level and/or $P(s)$ scaling, which is slightly different between samples. For example, it is expectable that higher noise level will lead to higher portion of intra-TAD interactions. This possibility should be excluded before drawing biological conclusions from very slight differences shown in Fig. 4 d and e. For example, note that in Fig. 4,E at the point -300 kb, the difference in insulation between POF and other stages is higher than in the TAD boundary region.

Response 1-2:

Thank you very much for these helpful comments. To effectively reduce the noise compared to intra-TAD interactions, we normalized TAD lengths by calculating TAD interaction enrichment and TAD intactness (Figure 4 e,f) in our revised manuscript. After normalization, the spatial segregation at the F1 stage is significantly higher than at the SWF stage ($P < 0.001$) and the POF stage ($P < 2.2 \times 10^{-16}$). Furthermore, the entropy status (VNE) of a *cis*-contact matrix was explored and confirms this result (Figure 2a,b).

As for the previous Figure 4e version, it seems normal that the difference in insulation between the POF stage and other stages is higher than in the TAD boundary region at the point -300 kb. When identifying the TAD boundary according to the insulation score (IS) algorithm, we normalized the IS of a bin relative to all bins across that chromosome using the following formula: $\log_2(\text{IS}/\text{mean_IS}(\text{all_bins}))$. Valleys/minima along the normalized IS vector represent loci of reduced Hi-C interactions across the bins and are considered TAD boundaries. As such, in our study, the median length of the TAD boundary is only 20 kb (at 20 kb resolution). The difference at the point ± 300 kb away from the center of the TAD boundary does not reflect the main characteristics of TADs, nor does it conflict with our conclusion about intra-TAD interactions.

Comment 1-3:

Promoter-enhancer interactions should be distinguished from CTCF-based loops formed via loop extrusion mechanism. It is known that CTCF sites are preferentially located near promoters and/or enhancers. Consistent with this, it seems that most of interactions depicted in Fig. 6, G and called by PSYCHIC connect TAD boundaries. These CTCF-mediated interactions might have regulatory effect; however, acute degradation of CTCF protein (Rao et al., 2017) showed that there are “bona fide” PEIs, which are mediated by mechanisms other than CTCF blocked extrusion. Authors should carefully distinguish these two types of interactions through the whole paragraph “Global remodeling of promoter-enhancer interactions in transcription control during follicle development”.

Response 1-3:

Thank you for this suggestion. We tried to construct CTCF ChIP-seq libraries for granular cells at the three stages but failed. It has been reported that PEI rewiring is often accompanied by changes in enhancer activity. Therefore, in the revised manuscript, we constructed six ChIP-seq libraries using an antibody against H3K27ac for GCs at the SWF, F1, and POF stages, with two biological duplications per stage, and annotated the genes contacting poised enhancers (PEs, 30.47%), regular enhancers (REs, 47.94%) and super-enhancers (SEs, 21.59%). These results accurately revealed the PEI regulatory networks in GCs during follicular development (Main text page 30, lines 497-507).

Comment 1-4:

“The spatial organization of compartments constructed by miniMDS³⁵ showed that PC1 value and gene expression are negatively associated with the distance from the center of the nucleus, i.e., the nuclear radius” – it is not clear how the nuclear radius was measured in each cell type.

Response: 1-4:

Thank you for this suggestion. We regret that the miniMDS³⁵ method was not

described in detail in the original manuscript version. In the revised manuscript, we added this part to the Methods section:

Materials and Methods

Chromatin 3D modeling and

The 3D chromosome conformations were inferred for each Hi-C map based on the normalized intra- (at 100 kb resolution) and inter- chromosomal (at 1-Mb resolution) contact maps using an approximation of multidimensional scaling (MDS) method as implemented in miniMDS program (Rieber and Mahony, 2017). After modeling, through Euclidean distance to measure the relative distance of each chromosome (100 kb resolution) to nucleolus (start point).

REF

Rieber, L. & Mahony, S. miniMDS: 3D structural inference from high-resolution Hi-C data. *Bioinformatics* 33, i261-i266, doi:10.1093/bioinformatics/btx271 (2017).

Comment 1-5:

The Introduction could be improved and better structured. In the current version Introduction ends with the too long section summarizing main results. By my opinion, major goal of the study is also not clearly stated in the Introduction.

Response: 1-5:

Thank you for this comment. We have simplified the summary (as shown in the following) and summarized the main goals in the **Introduction** section (Lines 62-72) in the revised manuscript.

Introduction

The domestic chicken (*Gallus gallus domesticus*), which includes broiler (meat-producing) and layer (egg-producing) chickens, is of enormous agricultural significance and represents a classic model to study folliculogenesis (Bahr et

al, .). In this study, we investigate the transcriptomic dynamics of GCs in ovarian follicles across ten key developmental stages and generate high-resolution chromatin contact maps for GCs across three major developmental stages using in situ high-throughput chromatin conformation capture (Hi-C) sequencing. These experimental settings allowed us to conduct an integrated analysis of chromatin structure and transcriptomic characterization of chicken GCs associated with various physiological functions during folliculogenesis.

Comment 1-6:

Please compare the RNAseq data obtained for chicken ovarian follicles with previously published studies describing chicken follicle transcriptome.

Response: 1-6:

Thanks for your comment. We have discussed the results in previously published RNA-seq studies focusing on chicken ovarian follicles in the **Discussion** section.

Discussion

Poultry breeders have always sought chickens with high egg production, and this trait depends on efficient ovarian development and ovulation. Additional insight into the gene transcription process during folliculogenesis will help to better understand the reproductive physiology of hens and eventually improve their laying performance. Here, we performed bulk RNA-seq to systematically investigate the gene expression profiles of GCs in ovarian follicles across the whole development process, including ten stages. Substantial transcriptomic dynamics showed distinct gene expression patterns corresponding to specific stages of ovarian follicle development. We found that the SWF, F1, and POF stages, which represent the prehierarchical, preovulatory, and postovulatory phases, respectively, had the largest transcriptome differences between each other among all stages of follicle development. GO terms of stage-specific signature genes of GCs were identified by scRNA-seq and emphasized the significant functional differences in the SWF, F1, and POF stages. These changes in gene expression, such as *AMH*, reflect the corresponding functional

characteristics of follicle development at different physiological stages, which was consistent with other RNA-seq studies in chicken follicle development (Zhu G et al., 2015; Zhu G et al., 2019).

Comment 1-7:

Please compare HiC data and identified TAD borders in chicken granulosa cells with other chicken cell types.

Response 1-7:

Thank you for your comment. To investigate the conservation of TAD in different cells, we downloaded chicken fibroblast and erithrocyte Hi-C data (including immature and mature erithrocytes) and identified TADs in these cells using methods similar to GCs. The comparison with TAD demonstrated that GCs and fibroblasts had TAD structures (mean Spearman's r of Direction Index = 0.87) that were more similar than GCs and erithrocyte cells ($r < 0.32$). Moreover, we identified 1996, 1361, and 1326 TAD boundaries in the chicken fibroblasts cells, immature and mature erithrocytes, of which, 82.01%, 55.84%, and 56.71% TAD boundaries are conserved in the GCs, respectively (Figure S7 d-e).

Comment 1-8:

By my opinion, the phrase «dramatic remodeling of genome structures» is too strong for the data obtained.

Response 1-8:

Thank you for your comment. We have revised the sentence to read “Notably, we found that higher-order chromatin structures, including compartmentalization, TAD boundaries, and intra-TAD interactions, were dynamic during the stage transformation of GCs and were associated with gene expression changes” (Main text page 37, lines 594-597).

Comment 1-9:

Within the Abstract: “we integrated RNA-seq and Hi-C analyses of chicken follicular granulosa cells of 10 developmental stages.” please correct the sentence so that it would be clear that HiC analyses was performed in 3 developmental stages.

Response 1-9:

Thank you for this correction. We have revised this sentence as suggested:

“We investigate the transcriptomic dynamics of chicken GCs over ten follicular stages and assess the chromatin architecture dynamics and how it influences gene expression in GCs at three key stages: the prehierarchical small white follicles (SWF), the first largest preovulatory follicles (F1), and the postovulatory follicles (POF).” (Main text page 1, lines 9-13).

Comment 1-10:

Fig 1b. Indicate in the figure legend what is shown by lines of different color.

Response 1-10:

Thanks for your comment. We have modified the figure legend for Figure 1b to read as follows: “The red lines represent mean gene expression levels, and the blue lines represent gene expression levels for each gene in the relative cluster during folliculogenesis.” We have also revised the legends throughout the manuscript.

Comment 1-11:

Fig 4h. Please show the HiC-contact heatmaps without TAD borders.

Response 1-11:

Thank you for this comment; we have modified the display of the Figure 4g.

Reviewer #2:

Comment 2-1:

The manuscript “Dynamic chromatin architecture in ovarian follicle development in Chickens” is an interesting and well written report. The authors integrated RNA-seq and Hi-C analyses of chicken follicular granulosa cells of 10 developmental stages. Overall, the authors identified novel mechanisms in chickens, such as spatial organization of compartments in the nucleus and correlation with gene expression, similar to what has been observed in mammals. The data is presented clearly and follows a logical path.

Response 2-1:

We are grateful for the reviewer’s help in reviewing our manuscript and for the positive feedback.

Comment 2-2:

One of the main concerns with this report is the analysis of the follicles as a whole unit, without the differentiation between the somatic and the germ compartments.

Selecting small white follicles (SWF), preovulatory (F1) and postovulatory follicles (POF) for the generation of genome-wide contact maps of chicken follicle development using Hi-C technology raises the question of how the authors compare SWF and F1, that contain an oocyte, with POFs that are devoid of oocytes. What is the contribution of the oocyte to the reported data?

In small follicles, I would assume that the oocyte contributes to 50% of the follicle. How authors separate the RNA contribution from the somatic versus germ cells to their data set?

Oocytes undergo significant changes during folliculogenesis, but the authors only reported data regarding granulosa cells without in-depth discussion of what is missing without the separate analysis of the oocytes.

The authors did not mention any limitations of the study. One limitation I could think about is the lack of the germ cell analysis.

Response 2-2:

We are grateful for your comments on our manuscript. In this study, we isolated the granulosa cells (GCs) in the follicle, which did not contain germ cells. These GCs were subsequently analyzed. We agree with the reviewer that our study is limited by a lack of germ cell analysis, which is addressed in the revised manuscript.

It is worth noting that we primarily assess GCs in this study. Advances in sequencing technology at single-cell resolution mean that further research is needed to explore the 3D genome architectural dynamics and how it influences gene expression in oocytes throughout the reproductive cycle, as well as analyze various epigenetic data to better understand regulatory mechanisms in follicle development.

Comment 2-3:

Abstract – please avoid using acronyms, such as SWF, F1, POF and TAD, in the abstract.

Response 2-3:

As suggested, we have deleted the unnecessary acronyms in both the abstract and the main text.

Reviewer #3:

Comment 3-1:

This work profiles the transcriptional dynamics of chicken follicular granulosa cells at ten developmental stages and chromatin folding dynamics using Hi-C at three critical stages (Small white follicle (SWF), growing preovulatory follicle stage F1 and postovulatory follicles (POF) stage SWF to POF using RNA-seq and characterised three stages (SWF, F1 and POF). Subsequently, the authors perform analyses including chromatin compartmentalisation, TADs and promoter-enhancer interactions and make a valid attempt towards integrating transcriptional and Hi-C data. Finally, the authors present two examples of local chromatin remodelling leading to expression upregulation during this

developmental process. Albeit descriptive, this study provides global profiling of chromatin structures in chicken ovarian follicle development and represents a worthy contribution to the field. The literature lacks comprehensive transcriptomics and chromatin folding analysis across the different developmental stages of the chicken ovarian follicles. This data will be beneficial for further future studies of gene regulatory processes in the ovarian follicles.

Response 3-1:

We appreciate the reviewer's comments and helpful suggestions for our manuscript.

Comment 3-2:

Line 312-317. The authors point out a strong correlation between changes in chromatin compartmentalisation but remark that the corresponding shifts in gene expression levels are missing. The main concern here is that this results from the coarse-grained resolution of chromatin compartmentalisation analysis currently presented in the paper. Ideally, authors need to increase the resolution by much deeper sequencing or perform targeted chromosome conformation capture experiments for several specific loci (4-C or Capture-C, for example) to address this issue. Furthermore, the authors should address this point in the discussion. The authors should comprehensively discuss to which extent interactions at various resolutions (e.g., A/B compartment, TAD and PEI) respectively contribute to changes in gene expression.

RNA-seq readout corresponds to mRNA levels. Both transcriptional activities and RNA degradation affect mRNA levels. The authors should discuss the relationships between changes in chromatin compartmentalisation and alterations in mRNA levels.

Response 3-2:

We appreciate the reviewer's valuable comments. In our study, we generated an ultra-deep Hi-C contact map at 5-kb resolution (~95.13% of 5 kb bins had at least 1,000 intrachromosomal contacts) by merging the Hi-C data of two

replicates. This enables us to identify PEIs at 5-kb resolution and investigate gene expression regulation (Page 29, Line 483). The deep Hi-C data can identify compartments and TADs at a 20-kb resolution, without merging contacts of replicates.

The correlation between chromatin conformation (form) and gene expression (function) can be complicated. A previously published Hi-C study revealed that compartments could only affect a subset of gene expression during stem cell differentiation (Dixon et al., 2015), which was similar to our findings. Given that open chromatin in the A compartment facilitates the binding of transcription factors and the formation of PEIs (Stevens et al., 2017), the regulatory role of compartments seems indirect and obscure. As the reviewer mentioned, the relatively weak correlation between changes in compartmentalization and alterations in mRNA levels could be due to factors like mRNA degradation. It has also been reported that the form-function relationship can be more accurately captured when comparing Hi-C data with nascent RNA data (e.g., bromouridine labeling and sequencing data), instead of with steady-state RNA-seq data.

TAD and chromatin loop can also participate in a gene regulatory programme. For example, Greenwald et al. (2019) reported that subtle changes in chromatin loop contact propensity are associated with differential gene regulation and expression. It has recently been found that the loop domain is better suited for cellular functions than compartment (Lu et al. 2020). Therefore, we conducted comprehensive analyses to depict changes in chromatin conformation at fine scales, including TAD boundary shifts, intra-TAD interaction changes, and PEI rewiring. We also explored their correlations with gene expression alterations. Further investigation is required to identify these interactions at various resolutions (e.g., A/B compartment, TAD, and PEI) and how they contribute to changes in gene expression.

REF

Dixon J R, Jung I, Selvaraj S, et al. Chromatin architecture reorganization during stem cell differentiation[J]. *Nature*, 2015, 518(7539): 331-336.

Greenwald WW, Li H, Benaglio P, Jakubosky D, Matsui H, Schmitt A, et al.

Subtle changes in chromatin loop contact propensity are associated with differential gene regulation and expression[J]. Nature Communications, 2019, 10: 1054.

Lu L, Liu X, Huang W K, et al. Robust Hi-C maps of enhancer-promoter interactions reveal the function of non-coding genome in neural development and diseases[J]. Molecular Cell, 2020, 79(3): 521-534. e15.

Stevens T J, Lando D, Basu S, et al. 3D structures of individual mammalian genomes studied by single-cell Hi-C[J]. Nature, 2017, 544(7648): 59-64.

Comment 3-3:

The authors should add a two-dimensional scatter plot to visualise the global relationships among RNA-seq samples.

Response3-3:

As suggested, we have added a scatter plot (Figure S1b) to the revised manuscript to display the relationship between the samples, which were obtained from the t-Distributed Stochastic Neighbor Embedding (t-SNE) analysis using bulk RNA-seq data.

Comment 3-4:

The authors have used maSigPro to cluster all genes to globally characterise transcriptomic changes and then used GO analysis to summarise the functions of genes assigned to each cluster. A well-known limitation of GO analysis is its over-generalisation and vagueness. Such an example is, for instance, the enrichment in the "growth and developmental processes" GO term. The authors should include a heatmap of enriched/representative genes that define particular terms between stages to replace fig 1d. Another option would be to use bubble plots to include both GO term enrichment and statistical significance values in the exact representation.

Response 3-4:

Thank you for your helpful comment. As suggested, the revised manuscript displays functional enrichment results in the following bubble plots: Figure 1c, Figure S1d, Figure S2f, Figure S9c, and Figure 6d in the revised version.

Comment 3-5:

Given the study's descriptive nature and lack of resolution in chromatin data, the authors should define specific active putative cis-regulatory elements by adding chromatin accessibility (ATAC-seq) characterisation at selected time points (same as Hi-C analysis). Integration of these data would greatly help understand the intricacies of the gene regulatory programme underlying chicken ovarian follicle development.

Response 3-5:

Thank you for your helpful comment. We have added the ATAC-seq analysis and included high-quality chromatin accessibility information in the revised manuscript. We also performed an ATAC-seq assay to measure differences in local accessibility during folliculogenesis. As expected, we found that the A compartments were enriched by more ATAC peaks than by their B compartments, making them more accessible. We observed that stage-specific peaks in GCs at the SWF and F1 stages are enriched in motifs corresponding to the transcription factors (TFs) in the GATA family, which are essential for the development, differentiation, and homeostasis processes. This suggests that the differentiation of GCs is highly active during these two stages (Aronson et al., 2014; Bertero et al., 2005). In contrast, POF-specific peaks in GCs are enriched in motifs corresponding to the TFs involved in cytotoxicity and the induction of apoptosis (Chae et al., 2005; Luo et al., 2021; Qin et al., 2018) (typically, KLF5, PITX1, and OTX1). As such, the chromatin accessibility statuses coincided with the identified A/B compartments based on Hi-C data and supported the physiological course of chicken folliculogenesis.

REF

Aronson, B. E., Stapleton, K. A. & Krasinski, S. D. Role of GATA factors in

development, differentiation, and homeostasis of the small intestinal epithelium. *Am J Physiol Gastrointest Liver Physiol* 306, G474-490, doi:10.1152/ajpgi.00119.2013 (2014).

Bertero, A. et al. Dynamics of genome reorganization during human cardiogenesis reveal an RBM20-dependent splicing factory. *Nature communications* 10, 1538, doi:10.1038/s41467-019-09483-5 (2019).

Chae, H. D. et al. Oocyte-based screening of cytokinesis inhibitors and identification of pectenotoxin-2 that induces Bim/Bax-mediated apoptosis in p53-deficient tumors. *Oncogene* 24, 4813-4819, doi:10.1038/sj.onc.1208640 (2005).

Luo, Y. & Chen, C. The roles and regulation of the KLF5 transcription factor in cancers. *Cancer Sci* 112, 2097-2117, doi:10.1111/cas.14910 (2021).

Qin, S. C. et al. Downregulation of OTX1 attenuates gastric cancer cell proliferation, migration and invasion. *Oncol Rep* 40, 1907-1916, doi:10.3892/or.2018.6596 (2018).

Comment 3-6:

It is surprising that using pairwise differential expression analysis, the authors found "no difference in gene expression between contiguous stages of development" (line 139-140), especially between SWF/LWF, LYF/F5 and F1/POF. Increasing the sequencing depth would likely for RNA-seq libraries as well allow identification of the differentially expressed genes between these stages (see comment above).

Response 3-6:

Thank you for this helpful comment. We increased the depth of sequencing (~13.54 Gb per library) and the number of samples (six replicates per time point) (Table S1) in the revised manuscript. As expected, 39~5,580 differentially expressed genes were identified between contiguous stages. We have updated this section in the revised version (page 4 lines 96-102).

Comment 3-7:

As admitted by the authors, "chromatin compartmentalisation contributes to relatively subtle changes in gene expression and does not play a deterministic role" (line 314-317). Hence, by definition, the significance of this work in terms of understanding the determinants of gene expression during follicle development is limited. One way to possibly improve the study's depth, bringing new insights, would be by extending the transcriptomic analysis using single-cell RNAseq on selected developmental stages - the study would then provide an atlas of both transcriptome and genome-wide chromatin interactions during chicken ovarian follicle development.

Response 3-7: Thank you for these suggestions. The compartmentalization reflects chromatin activity and is closely correlated with chromatin accessibility (Figure S6). Open chromatin could facilitate the binding of transcription factors and the formation of long-range chromatin contacts (e.g., loop domain) (Stevens et al., 2017). Chromatin compartmentalization contributes to changes in gene expression, which is described in our revised manuscript. Other hierarchical structures of chromatin conformation, including TAD and loops, also play important roles in regulating gene expression.

In the revised manuscript, we also generated six ChIP-seq libraries using H3K27ac antibodies and annotated the genes contacting poised enhancers (30.47%), regular enhancers (47.94%), and super-enhancers (21.59%). Investigating enhancer activity helps to reveal the PEI regulatory network during granulosa cell development (Main text page 30, lines 496-506).

While we did not generate follicular atlas single-cell data, we constructed single-cell libraries for granulosa cells (GCs) at the SWF, F1, and POF stages.

REF

Stevens T J, Lando D, Basu S, et al. 3D structures of individual mammalian genomes studied by single-cell Hi-C[J]. Nature, 2017, 544(7648): 59-64.

Comment 3-8:

Fig 1b legend. It is not clear what the lines denote. It is a very informative figure validating the temporal expression of representative genes from each cluster. First, the clusters can be ordered by cluster 2, 1, 4 and 3 to visualise the transition more clearly. Second, the vital genes of each cluster can be noted on the side of this heatmap.

Response 3-8:

Thanks for your comment. In Figure 1b of the revised manuscript, the red lines represent mean expression levels, and the blue lines represent each gene expression in a relative cluster during development. As suggested, we have rearranged the cluster order to more clearly visualize the transition (Figure 1b). Additionally, the genes in each cluster have been listed in Supplementary Table S3.

Comment 3-9:

Fig 2e. Authors should state the ratio of medians in addition to the p-value, as the low p-value can be mainly caused by a large number of sample sizes.

Response 3-9:

Thank you for your comment. We included the gene number and median values in Figure. S4d and other figures throughout the revised manuscript.

Comment 3-10:

The details of multiple testing correction and other statistical methods should be clearly noted in the text, methods and figure legends.

Response 3-10:

Thank you for your comment. As suggested, we included information about the statistical test in the figure legends and the **Materials and Methods** section (page 49, lines 944-947).

Materials and Methods

Statistical analyses

All statistical analyses were performed by Student's t-test or Wilcoxon rank-sum test using R. *, **, and *** in the figures were represent $P < 0.05$, $P < 0.01$, and $P < 0.001$, respectively.

Comment 3-11:

Page 16 Line 310 typo: "an transient transition."

Response 3-11:

We have carefully checked and corrected typographical and grammatical errors throughout the manuscript.

Reviewers' Comments:

Reviewer #2:

Remarks to the Author:

The research "Dynamic transcriptome and chromatin architecture in granulosa cells during chicken folliculogenesis" reported is an important contribution to the field. Most of questions and concerns from my previous review have been addressed. I recommend to accept the manuscript with some clarifications, which are listed below.

Response 1-2:

It is not clear how exactly new analysis solves the noise problem. Imagine TADs which have distance normalized bona fide (i.e. with zero noise) contact frequency e_{tads} , and intra-TADs having bona fide distance-normalized contact frequency e_{intra} .

Now if we consider samples with different noise levels e_{noise} this will results in observed interactions

$(e_{tads} + e_{noise})$ and $(e_{intra} + e_{noise})$

Could you convince the readers that changing the parameter e_{noise} over fixed parameters e_{tads} and e_{intra} will not affect the results of computations performed using TAD intactness of VNE math? It seems that at least TAD intactness, which is essentially ratio

$(e_{tads} + e_{noise}) / (e_{intra} + e_{noise})$

will depend on e_{noise} value, and I assume that VNE will behave similarly.

Response 1-3:

In fact, CTCF peaks are associated with enhancer and H3K27ac signal as well, thus H3K27ac annotation can not exclude the possibility of CTCF-mediated interactions. If annotating CTCF-mediated interactions is not possible, I would recommend to highlight that some PEIs might be actually CTCF-mediated loops in results and discussion sections.

Reviewer #3:

Remarks to the Author:

The addition of Figure 1b-c and the discussion of different clusters significantly improved the clarity of the manuscript. The authors addressed the reviewer's concerns regarding the impurity and unintended inclusion of germ cells in the analysis. Below are a few minor typos and grammar mistakes that need to be addressed before publication.

Abstract:

1. There is a typo in Line 13: "Our results provide demonstrate" – please delete the word "provide".
2. Line 15-16: Please consider replacing the word "including" with "for": "providing ample evidence for compartmentalization ..."
3. Last sentence in the abstract: the authors state that their results "lay the ground work for in-depth functional characterization", but it is not clear functional characterization of what?

Introduction:

1. Line 27-28: I think the more accurate statement would be: "...consists of follicles at several different developmental stages", not several follicles.
2. Lines 53-54: Please, check the grammar of the statement: "... where there is a focus on the activation of primordial follicles or growth follicle selection". What do you mean by "growth follicle selection"
3. Line 57: Typo in the word "interactions"

Results:

Lines 278-280: The paragraph ends with a sentence "These results support 279 the physiological course of chicken folliculogenesis." and the next paragraph starts with a similar sentence "These results provide evidence...."

I am confused to which results authors refer to in the new paragraph starting on Line 280.

Figure 1a: I suggest adding "Number of follicles" in the schematic itself in addition to the mention

in the figure legend.

Supplementary Figure 2,4,5,6,7 and 8 legend: typo: granulosa not granule cells

Figure 2,3,4 and 5 legend: typo: granulosa not granule cells

Reviewer #5:

Remarks to the Author:

The authors have substantially revised the manuscript entitled "Dynamic transcriptome and chromatin architecture in granulosa cells during chicken folliculogenesis" according to the reviewer's (Reviewer #3) earlier comments.

The substantial revision made by authors include: added a scatter plot to visualize the global relationships among RNA-seq samples; added several bubble plots to better display both gene ontology (GO) term enrichment and statistical significance values in the exact representation; added the ATAC-seq analysis and included high-quality chromatin accessibility information; increased the depth of sequencing and the number of samples to identify DEGs between contiguous stages; generated ChIP-seq libraries using H3K27ac antibodies and investigated the enhancers (such as poised enhancers, regular enhancers, and super-enhancers) activity to reveal the promoter-enhancer interaction (PEI) regulatory network during granulosa cell development; added scRNA-seq data of granulosa cells isolated at three representative stages (SWF, F1, and POF); and improved the figure and supplementary table showing temporal expression of genes from four cluster during folliculogenesis. For some comments, particularly in comment 3-2, the authors gave sensible answers and reflected that in the revised discussion.

Overall, the revised manuscript and authors point-by-point response to Reviewer #3's earlier comments are satisfactory.

However, several additional minor issues should be corrected in the revised manuscript.

Minor comments

Please correct the phrases at the following sentences: Page 1, line 13. "Our results provide demonstrate the"; Page 2, line 54. "growth follicle selection"

Page 4, line 98. "of the three prehierarchical stages". Is this three or four?

Page 5, line 129. "post-ovulatory POF stage". Write this stage correctly (as POF stage) here and several other places.

Supplementary Fig. 1b and 1e was cited again when describing the results of scRNA-seq (line 129). Citation here is not necessary and could leads to misunderstand the results from different techniques.

"chicken granule cells" should be "chicken granulosa cells" at line 176 and many other places.

Page 5, lines 132-136. It is not clear why the chicken heart cells were used as comparative controls for scRNA-seq of GCs. Please write few lines of reason. Also, add 2-3 marker expression to confirm the identity of heart cells in suppl. fig. 2c.

Similar to above comment, explanation is needed for why the chicken fibroblasts cells and erythrocytes were used for the comparison of TADs in GCs. Correct the typo as "erythrocytes" here and several other places. Add the definition of CEF, CIME, and CME.

Please cite the figure panels in a serial manner. Supplementary Fig. 11e appears at line 499, but the panels a-d appears at line 532.

Detailed responses to reviewers

All comments provided by reviewers are given in gray italics, and our responses are in black. All revisions in the manuscript are marked in red.

Reviewer #1

Comment 1-1

The research "Dynamic transcriptome and chromatin architecture in granulosa cells during chicken folliculogenesis" reported is an important contribution to the field. Most of questions and concerns from my previous review have been addressed. I recommend to accept the manuscript with some clarifications, which are listed below.

It is not clear how exactly new analysis solves the noise problem. Imagine TADs which have distance normalized bona fide (i.e. with zero noise) contact frequency e_{tads} , and intra-TADs having bona fide distance-normalized contact frequency e_{intra} . Now if we consider samples with different noise levels e_{noise} this will results in observed interactions $(e_{tads} + e_{noise})$ and $(e_{intra} + e_{noise})$.

Could you convince the readers that changing the parameter e_{noise} over fixed parameters e_{tads} and e_{intra} will not affect the results of computations performed using TAD intactness of VNE math? It seems that at least TAD intactness, which is essentially ratio $(e_{tads} + e_{noise}) / (e_{intra} + e_{noise})$ will depend on e_{noise} value, and I assume that VNE will behave similarly.

Response 1-1

We appreciate this thoughtful comment. To investigate the robustness of the TAD intactness measurement, we generated the simulated contact matrix by randomly adding signal noises to different proportions (from 5% to 50%) of the contact matrix at different noise signal levels (from 10% to 100% of the average contact frequency at each distance in the original contact matrix).

As shown in **Figure S8 (newly added, Main text pages 26-27, lines 421-443)**,

we found the TAD intactness was highly robust across variable noise levels. Even at noise levels as high as 60% for 50% of the contact matrix, the TAD intactness between the original and simulated contact matrix is approximately equal (fold change of intactness = 1, $P = 0.65$, Wilcoxon rank-sum test).

Comment 1-2

In fact, CTCF peaks are associated with enhancer and H3K27ac signal as well, thus H3K27ac annotation can not exclude the possibility of CTCF-mediated interactions. If annotating CTCF-mediated interactions is not possible, I would recommend to highlight that some PEIs might be actually CTCF-mediated loops in results and discussion sections.

Response 1-2

Thank you for your constructive comments regarding the annotation of PEIs mediated by CTCF loops. Per your suggestion, we used the consensus CTCF motif of vertebrates to *in silico* predict the CTCF motif loci and their orientations in the chicken genome. As expected, these CTCFs were enriched in TAD boundaries and enhancer regions (**newly added Figure S12k**).

We also identified loops in SWF, F1, and POF at 5 kb resolution in the genomic distance range of 20 kb to 2 Mb using the Fit-Hi-C Python package. We next applied a hard cutoff to obtain the top 15,000 loops by ranking the strength of the loops. Finally, we obtained 6,632, 7,050, and 8,023 convergent CTCF-CTCF loops in SWF, F1, and POF, respectively. The genes in these loops have 4,067, 4,926 and 2,582 PEIs respectively. As expected, the higher proportion of these PEI events (76.52%, 78.56%, and 80.38% for SWF, F1, and POF respectively) have occurred within loops (**newly added Figure S12l**).

We have added a Supplementary table for these annotated convergent CTCF-CTCF loops and relative PEI information (**Table S9**), as well as an annotation of the CTCF loop in **Fig. 6e (Main text page 33, lines 566-568)**.

Newly added descriptions in Methods (**Main text page 47, lines 866-877**):

Identification of convergent CTCF-CTCF loops

We used the FIMO software (v5.1.1) to identify CTCF motif loci and their orientations in the GRCg6a version of the chicken genome based on consensus CTCF motif from the JASPAR CORE 2016 vertebrate database. As expected, these CTCFs were enriched in TAD boundaries and enhancer regions (**Figure S12k**). We also separately identified loops in SWF, F1, and POF at 5 kb resolution in the genomic distance range of 20 kb to 2 Mb using the Fit-Hi-C Python package (v2.0.7) (q value < 0.05). To further obtain the highly confident loops, we applied a hard cutoff to obtain the top 15,000 loops by ranking the loop strengths. We finally obtained 6,632, 7,050, and 8,023 convergent CTCF-CTCF loops in SWF, F1, and POF, respectively. The genes in these loops have 4,067, 4,926 and 2,582 PEIs respectively.

Newly added descriptions in Results:

In addition, the convergent CTCF-CTCF loop associated with FDX1 was only identified in the F1 stage, which is consistent with the most PEIs detected in this stage (**Fig. 6e**). This result indicates that CTCF sites are preferentially located near promoters and/or enhancers to constrain the PEIs (**Supplementary Fig. 12k-l, Supplementary Table 9**).

Reviewer #2

Comment 2-1

The addition of Figure 1b-c and the discussion of different clusters significantly improved the clarity of the manuscript. The authors addressed the reviewer's concerns regarding the impurity and unintended inclusion of germ cells in the analysis. Below are a few minor typos and grammar mistakes that need to be addressed before publication.

Response 2-1

We appreciate your comments.

Comment 2-2

Abstract:

1. *There is a typo in Line 13: “Our results provide demonstrate” – please delete the word “provide”.*

Response 2-2

Corrected as suggested.

Comment 2-3

2. *Line 15-16: Please consider replacing the word “including” with “for”: “providing ample evidence for compartmentalization ...”*

Response 2-3

Corrected as suggested.

Comment 2-4

3. *Last sentence in the abstract: the authors state that their results “lay the ground work for in-depth functional characterization”, but it is not clear functional characterization of what?*

Response 2-4

As suggested, the sentence was rewritten to “These results provide key insights into avian reproductive biology and provide a foundational dataset for the future in-depth functional characterization of GCs.” (**Main text page 1, lines 18-20**).

Comment 2-5

Introduction:

1. Line 27-28: I think the more accurate statement would be: "...consists of follicles at several different developmental stages", not several follicles.

Response 2-5

Yes. We agree with your comments. As suggested, the sentence was rewritten to "...consists of follicles at several different developmental stages" (**Main text page 1, line 27**).

Comment 2-6

2. Lines 53-54: Please, check the grammar of the statement: "... where there is a focus on the activation of primordial follicles or growth follicle selection". What do you mean by "growth follicle selection"

Response 2-6

We revised "growth follicle selection" to "selection of a dominant follicle" (**Main text page 2, line 52**).

Comment 2-7

3. Line 57: Typo in the word "interactions"

Response 2-7

Corrected as suggested.

Comment 2-8

Results:

Lines 278-280: The paragraph ends with a sentence "These results support the physiological course of chicken folliculogenesis." and the next paragraph starts with a similar sentence "These results provide evidence...."

I am confused to which results authors refer to in the new paragraph starting on Line 280.

Response 2-8

Thank you for this thoughtful comment. In the revised manuscript, we rewritten as below:

“These results support the physiological course of chicken folliculogenesis, provide evidence that chromatin state-mediated compartment activation is associated with transcriptional regulation, and directly implicate multiple loci that exhibited distinct compartmentalization and accessibility during folliculogenesis” (**Main text page 14, lines 281-285**).

Comment 2-9

Figure 1a: I suggest adding “Number of follicles” in the schematic itself in addition to the mention in the figure legend.

Response 2-9

Done as suggested. We have added “Number of follicles” in the schematic (**Figure 1a**).

Comment 2-10

Supplementary Figure 2,4,5,6,7 and 8 legend: typo: granulosa not granule cells

Figure 2,3,4 and 5 legend: typo: granulosa not granule cells

Response 2-10

Corrected as suggested.

Reviewer #4

Comment 4-1

The authors have substantially revised the manuscript entitled “Dynamic transcriptome and chromatin architecture in granulosa cells during chicken

folliculogenesis” according to the reviewer’s (Reviewer #3) earlier comments.

The substantial revision made by authors include: added a scatter plot to visualize the global relationships among RNA-seq samples; added several bubble plots to better display both gene ontology (GO) term enrichment and statistical significance values in the exact representation; added the ATAC-seq analysis and included high-quality chromatin accessibility information; increased the depth of sequencing and the number of samples to identify DEGs between contiguous stages; generated ChIP-seq libraries using H3K27ac antibodies and investigated the enhancers (such as poised enhancers, regular enhancers, and super-enhancers) activity to reveal the promoter-enhancer interaction (PEI) regulatory network during granulosa cell development; added scRNA-seq data of granulosa cells isolated at three representative stages (SWF, F1, and POF); and improved the figure and supplementary table showing temporal expression of genes from four cluster during folliculogenesis. For some comments, particularly in comment 3-2, the authors gave sensible answers and reflected that in the revised discussion.

Overall, the revised manuscript and authors point-by-point response to Reviewer #3’s earlier comments are satisfactory.

However, several additional minor issues should be corrected in the revised manuscript.

Response 4-1:

We highly appreciate your positive comments.

Comment 4-2

Minor comments

Please correct the phrases at the following sentences: Page 1, line 13. “Our results provide demonstrate the”;

Response 4-2

Sorry for this careless mistake. Corrected as suggested. We have also gone through the whole manuscript carefully and revised the phrases.

Comment 4-3

Page 2, line 54. "growth follicle selection"

Response 4-3

"growth follicle selection" was replaced by "the selection of a dominant follicle"

Comment 4-4

Page 4, line 98. "of the three prehierarchical stages". Is this three or four?

Response 4-4

It is four: "of the three prehierarchical stages" was changed to "of the four prehierarchical stages".

Comment 4-5

Page 5, line 129. "post-ovulatory POF stage". Write this stage correctly (as POF stage) here and several other places.

Response 4-5

Thank you. "the post-ovulatory POF stage" was replaced by "the POF stage" in our revised manuscript.

Comment 4-6

Supplementary Fig. 1b and 1e was cited again when describing the results of scRNA-seq (line 129). Citation here is not necessary and could leads to misunderstand the results from different techniques.

Response 4-6

Thank you, we agree with this comment. The citation of **Supplementary Fig. 1b** and **1e** in the results of scRNA-seq was removed as suggested.

Comment 4-7

“chicken granule cells” should be “chicken granulosa cells” at line 176 and many other places.

Response 4-7

This has been corrected.

Comment 4-8

Page 5, lines 132-136. It is not clear why the chicken heart cells were used as comparative controls for scRNA-seq of GCs. Please write few lines of reason. Also, add 2-3 marker expression to confirm the identity of heart cells in suppl. fig. 2c.

Response 4-8

Thank you for your valuable comments. According to your suggestion, we added below statements in the main text (**Main text page 5, lines 131-133**):

“To confirm the identity of these GCs, we used publicly available transcriptome profiles of 22,561 cells derived from chicken hearts as comparative controls.”

As per your suggestions, to confirm the identity of heart cells, we also provided the expression of five marker genes of heart cells in chicken (*MSX1*, *HBZ*, *FABP5*, *LCP1* and *ALDH1A2*) (Madhav Mantri et al., 2021) in **suppl. Fig. 2c**.

Comment 4-9

Similar to above comment, explanation is needed for why the chicken fibroblasts cells and erythrocytes were used for the comparison of TADs in

GCs.

Response 4-9

Thank you for your concern. According your suggestions, we added below statements in the main text (**Main text page 21, lines 356-358**):

“To investigate the conservation of TAD in different cells, we downloaded chicken fibroblast and erythrocytes Hi-C data (including immature and mature erythrocytes) and performed a comparative analysis”.

Comment 4-10

Correct the typo as “erythrocytes” here and several other places.

Response 4-10

Corrected as suggested.

Comment 4-11

Add the definition of CEF, CIME, and CME.

Response 4-11

Done as suggested.

Comment 4-12

Please cite the figure panels in a serial manner. Supplementary Fig. 11e appears at line 499, but the panels a-d appears at line 532.

Response 4-12

The figure panels have been carefully checked and cited in our revised manuscript.

Reviewers' Comments:

Reviewer #2:

Remarks to the Author:

The research "Dynamic transcriptome and chromatin architecture in granulosa cells during chicken folliculogenesis" reported is an important contribution to the field. Most of questions and concerns from my previous review have been addressed. I thus happy to recommend to accept the manuscript.